# Encoded Prior Sliced Wasserstein AutoEncoder for Learning Latent Manifold Representations

## Abstract

While variational autoencoders have been successful in a variety of tasks, the use of conventional Gaussian or Gaussian mixture priors are limited in their ability to encode underlying structure of data in the latent representation. In this work, we introduce an Encoded Prior Sliced Wasserstein AutoEncoder (EPSWAE) wherein an additional prior-encoder network facilitates learns an embedding of the data manifold which preserves topological and geometric properties of the data, thus improving the structure of latent space. The autoencoder and prior-encoder networks are iteratively trained using the Sliced Wasserstein (SW) distance, which efficiently measures the distance between two *arbitrary* sampleable distributions without being constrained to a specific form as in the KL divergence, and without requiring expensive adversarial training. To improve the representation, we use (1) a structural consistency term in the loss that encourages isometry between feature space and latent space and (2) a nonlinear variant of the SW distance which averages over random nonlinear shearing. The effectiveness of the learned manifold encoding is best explored by traversing the latent space through interpolations along *geodesics* which generate samples that lie on the manifold and hence are advantageous compared to standard Euclidean interpolation. To this end, we introduce a graph-based algorithm for interpolating along network-geodesics in latent space by maximizing the density of samples along the path while minimizing total energy. We use the 3D-spiral data to show that the prior does indeed encode the geometry underlying the data and to demonstrate the advantages of the network-algorithm for interpolation. Additionally, we apply our framework to MNIST, and CelebA datasets, and show that outlier generations, latent representations, and geodesic interpolations are comparable to the state of the art.

## 1 Introduction

Generative models have the potential to capture rich representations of data and use them to generate realistic outputs. In particular, Variational AutoEncoders (VAEs) (Kingma & Welling, 2014) can capture important properties of high-dimensional data in their latent embeddings, and sample from a prior distribution to generate realistic images. Whille VAEs have been very successful in a variety of tasks, the use of a simplistic standard normal prior is known to cause problems such as under-fitting and over-regularization, and fails to use the network's entire modeling capacity (Burda et al., 2016). Gaussian or Gaussian mixture model (GMM) priors are also limited in their ability to represent geometric and topological properties of the underlying data manifold. High-dimensional data can typically be modeled as lying on or near an embedded low-dimensional, nonlinear manifold (Fefferman et al., 2016). Learning improved latent representations of this nonlinear manifold is an important problem, for which a more flexible prior may be desirable.

Conventional variational inference uses Kullback-Leibler (KL) divergence as a measure of distance between the posterior and the prior, restricting the prior distribution to cases that have tractable approximations of the KL divergence. Many works such as Guo et al. (2020); Tomczak & Welling (2018); Rezende & Mohamed (2015) etc. have investigated the use of more complicated priors (notably GMMs) which lead to improved latent representation and generation compared to a single Gaussion prior. Alternate approaches such as adversarial training learn arbitrary priors by using a

discriminator network to compute a divergence (Wang et al., 2020), however they are reported to be harder to train and are computationally expensive.

In this work, we introduce the Encoded Prior Sliced Wasserstein AutoEncoder (EPSWAE), which consists of a conventional autoencoder architecture and an additional prior-encoder network that learns an unconstrained prior distribution that encodes the geometry and topology of *any* data manifold. We use a type of Sliced Wasserstein (SW) distance (Bonnotte, 2013; Bonneel et al., 2015), a concept from optimal transport theory that is a simple and convenient alternative to the KL divergence for any sampleable distributions. A Sliced Wasserstein AutoEncoder (SWAE) that regularizes an autoencoder using SW distance was proposed in Kolouri et al. (2018a). Several works improve the SW distance through additional optimizations (Deshpande et al., 2019; Chen et al., 2020b; Deshpande et al., 2018), and show improved generation, however involve additional training and use a fixed (usually Gaussian) prior. Kolouri et al. (2019) presents a comparison between max-SW distance, polynomial generalized SW distances, and their combinations. In contrast, we use a simple and efficient nonlinear shearing which requires no additional optimization.

Additionally, we introduce a structural consistency term that encourages the latent space to be isometric to the feature space, which is typically measured at the output of the convolutional layers of the data encoder. Variants of this penalty have previously been used to encourage isometry between the latent space and data space (Yu et al., 2013; Benaim & Wolf, 2017; Sainburg et al., 2018). The structural consistency term further encourages the prior to match the encoded data manifold by preserving *feature-isometry*, which in turn is expected to assist with encoding the geometry of the data manifold , thus leading to improved latent representations.

A key contribution of our work is the graph-based geodesic-interpolation algorithm. Conventionally, VAEs use Euclidean interpolation between two points in latent space. However, since manifolds typically have curvature, this is an unintuitive distance metric that can lead to unrealistic intermediate points. Our goal is to learn a true representation of the underlying data manifold, hence it is natural to interpolate along the manifold geodesics in latent space. Several works such as Shao et al. (2018); Miolane & Holmes (2020b) endow the latent space with a Riemannian geometry and measure corresponding distances, however these are difficult and involve explicitly solving expensive ordinary differential equations. In this work, we introduce 'network-geodesics', a graph-based method for interpolating along a manifold in latent space, that maximizes sample density along paths while minimizing total energy. This involves first generating a distance graph between samples from the prior. Then this network is non-uniformly thresholded such that the set of allowable paths from a given sample traverse high density regions through short hops. Lastly, we use a shortest path algorithm like Dijkstra's algorithm (Dijkstra, 1959) to identify the lowest 'energy' path between two samples through the allowable paths. Since the prior is trained to learn the data manifold, the resulting network-geodesic curves give a notion of distance on the manifold and can be used to generate realistic interpolation points with relatively few prior samples.

The novel contributions of this work are:

- We introduce a novel architecture, EPSWAE, that consists of a prior-encoder network that is efficiently trained (without expensive adversarial methods) to generate a prior that encodes the geometric and topological properties of data.
- We introduce a novel graph-based method for interpolating along network-geodesics in latent space through maximizing sample density while minimizing total energy. We show that it generates natural interpolations through realistic images.
- Improvements to the latent space representation are obtained by using a structural consistency term in the loss that encourages isometry between feature space and latent space and by using a simple and efficient nonlinear variant of the SW distance.

## 2 BACKGROUND AND RELATED WORK

Several works have attempted to increase the complexity of the prior in order to obtain better latent representations. Most data can mathematically be thought of as living on a high dimensional manifold. In an image dataset, for instance, if images in high dimensional pixel space are effectively parametrized using a small number of continuous variables, they will lie on or near a low dimensional manifold (Lu et al., 1998). Many works such as Weinberger & Saul (2006); Rahimi et al.

(2005) have investigated image and video manifolds. Encoding and exploiting the topological and geometric properties of such data is a question of increasing interest.

Some representative works are Dilokthanakul et al. (2016) which shows improved unsupervised clustering of latent space by using GMM priors, Takahashi et al. (2019) which uses the density ratio trick to calculate the KL divergence using implicit priors, Rezende & Mohamed (2015), which maps a Gaussian prior by a series of explicit transformations, Guo et al. (2020) which learns a GMM prior using an approximate ELBO, Goyal et al. (2017) which uses a hierarchical Bayesian framework, and VampPrior Tomczak & Welling (2018) which learns a two-layer hierarchical GMM from aggregated posteriors. Yin & Zhou (2018); Molchanov et al. (2019); Liutkus et al. (2019) expand the variational family to incorporate implicit variational distributions while providing exact theoretical support.

The KL divergence is tractable only for Gaussian distributions, however the Wasserstein distance of 1D projections (also called Sliced Wasserstein (SW) distance) has a closed-form for any arbitrary distribution (Kolouri et al., 2018b). Wasserstein distances (Villani, 2003) have several of the same properties as KL divergence, but often lead to better sampling (Gulrajani et al., 2017; Tolstikhin et al., 2018), and have been used in several machine learning applications (e.g. Arjovsky et al. (2017); Tolstikhin et al. (2018); Kolouri et al. (2018b)).

A Sliced Wasserstein AutoEncoder (SWAE) that regularizes an autoencoder using SW distance was proposed in Kolouri et al. (2018a). There exist several variants of SW distance that have been successful at improving generative ability - for example, Deshpande et al. (2018); Chen et al. (2020b) train discriminator-like networks to perform nonlinear transformations (instead of purely linear projections) whereas Deshpande et al. (2019) introduces max-SW distance, which looks for the best linear projection, generalized in Nguyen et al. (2020) to finding optimal distributions of projections. Wu et al. (2019) uses a different optimization approach to computing the SW distance based on the Kantorovich dual formulation. All of the above use some form of additional training. In contrast, we use a simple and efficient nonlinear shearing which requires no additional optimization and is sufficient for improving latent representation, however, our method can easily incorporate other SW distances if desired.

Alternatively, many works such as Wang et al. (2020); Makhzani et al. (2015); Arjovsky et al. (2017); Sainburg et al. (2018) replace the KL divergence with an adversarial loss, however adversarial methods tend to be significantly more expensive and difficult to optimize. In higher dimensions, using a discriminator network as in adversarial approaches is a natural way of implicitly computing an equivalent of the Wasserstein-1 distance (Arjovsky et al., 2017; Tolstikhin et al., 2018). However, using the SW distance is much simpler and more efficient. Adversarial training for interpolation in Sainburg et al. (2018) uses a structural consistency term that encourages relative distances in data space to be preserved in latent space. Similar distance-preserving terms have also been used successfully in Yu et al. (2013); Benaim & Wolf (2017), however Euclidean distances in data space can be a poor measure of the natural geometry of the data. In our work, we preserve relative distances in latent space and *feature* space to improve latent encoding, where features are extracted at the output of the last convolutional layer in the data encoder.

Several manifold learning techniques such as Dollár et al. (2007); Weinberger & Saul (2006) compute embeddings of high-dimensional data but are less general and lack generative or interpolation abilities. The hyperspherical VAE Davidson et al. (2018) outperforms the standard VAE for data residing on a hyperspherical manifold. Miolane & Holmes (2020a) formulates a Riemannian VAE, however computing its ELBO is challenging. Along similar lines, Arvanitidis et al. (2017) shows that under certain conditions, a Riemannian metric is naturally induced in the latent space, and uses it to compute geodesics. In Chen et al. (2020a), a flat manifold is approximated by penalizing curvature in latent space, and geodesics are defined through Euclidean distances on the flat manifold.

There exists limited work on integrating graphical structures with generative models (Kipf & Welling, 2016). Hadjeres et al. (2017) studies monophonic music using a latent space geodesic regularization that allows interpolations in the latent space to be more meaningful, giving a notion of geodesic distance. Several approaches such as Tenenbaum et al. (2000); Bernstein et al. (2000); Klein & Zachmann (2004); Mémoli & Sapiro (2005); Luo & Hu (2020) approximate geodesics on point clouds by, for instance, building a nearest-neighbor network on the manifold and applying a shortest path algorithm, however, these often require many samples in order generate reasonable approximations and aren't robust to noise (Sober et al., 2020). Inspired by this, we introduce an

energy-based network algorithm to identify network-geodesics in latent space with relatively few, noisy samples.

## 3 EPSWAE

### 3.1 NONLINEAR SLICED WASSERSTEIN DISTANCE

Wasserstein distances are a natural metric for measuring distances between probability distributions, however they xare difficult to compute in dimensions two and higher. The Sliced Wasserstein distance $d_{SW}$ averages over the Wasserstein distance of 1D projections (see Appendix A for derivation). Several works (e.g. Kolouri et al. (2019); Deshpande et al. (2019)) discuss why a conventional SW distance may be sub-optimal as a large number of linear projections may be required to distinguish two distributions. In this work we use a Nonlinear Sliced Wasserstein distance ($d_{NSW}$), an averaging procedure over (random) *nonlinear* 1D projections, between two distributions $\mu$ and $\nu$ defined as:

$$d_{NSW}(\mu, \nu) = \mathbb{E}d_{SW}(\mathcal{N}_*^{\zeta,\gamma}\mu, \mathcal{N}_*^{\zeta,\gamma}\nu) \approx \frac{1}{L}\sum_{\ell=1}^{L} d_{SW}(\mathcal{N}_*^{\zeta_\ell,\gamma_\ell}\mu, \mathcal{N}_*^{\zeta_\ell,\gamma_\ell}\nu) \qquad (1)$$

where $L$ is the number of nonlinear transformations, and $\zeta, \gamma$ are chosen to be normal random variables with mean 0, and variance matching that of $\mu$, and $\mathcal{N}$ is defined below. We define the "push-forward" (also known as "random variable transform") as follows: given any function $f$ : $X \to Z$ and probability measure $\mu$, we define $f_*\mu(A) = \mu(f^{-1}(A))$ for all $A \subset Z$ measurable. Most importantly, to generate samples of $z \sim f_*\mu$ one simply generates $x \sim \mu$ and defines $z = f(x)$.

The nonlinear transformations are related to a special case of the generalized SW distance (Kolouri et al., 2019) (with the difference that we choose distribution-dependent shear frequencies that are most likely to produce deformations which highlight differences in the measures, thus breaking the homogeneity condition H2 in Kolouri et al. (2019)), and are given by

$$\mathcal{N}^{\zeta_\ell,\gamma_\ell}(x) = x + \zeta_\ell \sin(\gamma_\ell \cdot x). \qquad (2)$$

In all our experiments, we use $L = 5$ nonlinear transformations with 50 linear 1D projections each. Several works such as Deshpande et al. (2018; 2019); Wu et al. (2019); Nguyen et al. (2020); Chen et al. (2020b) improve the SW distance through some form of optimization or training an additional discriminator-like network, and show improved generative results (with Deshpande et al. (2018) being the most similar to our method). We did not implement these other variants, however they could easily be used in conjunction with our method. Our choice of nonlinearity is motivated by certain tail-behavior considerations (such as boundedness and non-saturation of nonlinearity), along with computational efficiency, and for our goals, a simple nonlinearity was sufficient. See section 4.2 and Appendix E for computational time and loss comparisons with some other nonlinearities discussed in Kolouri et al. (2019). For further discussion on NSW distance and derivation see Appendix A.

### 3.2 ALGORITHM DETAILS

In the Encoded Prior Sliced Wasserstein AutoEncoder (EPSWAE) (see Fig. 1 for schematic), let's define the data encoder as $\Psi_E$, the decoder $\Psi_D$, and the prior-encoder as $\Psi_{PE}$, each with parameters $\phi_E, \phi_D, \phi_{PE}$ respectively. Input samples $\mathbf{x}^{(j)} \sim P_X$, where $P_X$ is the probability distribution of the input data, are passed through $\Psi_E$ to generate posterior samples $\mathbf{z}^{(j)} \sim (\Psi_E)_* P_X$ by setting $\mathbf{z}^{(j)} = \Psi_E(\mathbf{x}^{(j)})$. Similarly, prior-encoder input samples $\xi^{(j)} \sim \mu$, where $\mu$ is the probability distribution of the input to the prior-encoder (chosen to be a mixture of Gaussians) from a distribution, are passed through $\Psi_{PE}$ to generate prior samples $\mathbf{y}^{(j)} \sim (\Psi_{PE})_* \mu$ by setting $\mathbf{y}^{(j)} = \Psi_{PE}(\xi^{(j)})$. The prior-encoder network and the autoencoder (data encoder and decoder) network are trained iteratively in a two step process:

1. Given a minibatch, parameters of the autoencoder ($\phi_E, \phi_D$) are trained for $k_1$ steps while parameters of the prior encoder ($\phi_{PE}$) are fixed. The loss function for the autoencoder consists of the reconstruction error, the NSW distance, and a Feature Structural Consistency (FSC) term $\mathcal{L}_{FSC}$ (given in Eqn. 4):

$$\mathcal{L}_{AE} = \alpha\mathbb{E}_{\mathbf{x}\sim P_X}\mathcal{L}_{rec}(\mathbf{x}, \bar{\mathbf{x}}) + \beta d_{NSW}((\Psi_E)_* P_X, (\Psi_{PE})_*\mu) + \kappa\mathcal{L}_{FSC}, \qquad (3)$$

where $\bar{\mathbf{x}} = \Psi_D(\Psi_E(\mathbf{x}))$ is the autoencoder output. Note that $d_{NSW}$ is efficiently computed just from samples of the distributions.

In this measure theoretic notation, $(\Psi_E)_* P_X$ is the posterior distribution (denoted as $q_{\phi_E}(z|x)$ in Bayesian literature), and $(\Psi_{PE})_* \mu$ is the prior distribution (denoted as $p_{\phi_{PE}}(z)$ in Bayesian literature). Similar to $\beta$-VAE ((Higgins et al., 2016)), the hyperparameters $\alpha, \beta, \kappa$ are tuneable; in our experiments they are fixed, but it could be advantageous to have them vary over training epochs.

$\mathcal{L}_{FSC}$ encourages relative distances in feature space to be preserved in latent space, where features are extracted at the output of the last convolutional layer in the encoder (or at input data if no convolutional layers are present). $\mathcal{L}_{FSC}$ (adapted from Sainburg et al. (2018)) for two point clouds $F = [f_1, \ldots f_N]$ in feature space and the corresponding points $Z = [z_1, \ldots z_N]$ in latent space in a minibatch of size $N$ is given by:

$$\mathcal{L}_{FSC} = \frac{1}{N^2} \sum_{\ell,j=1}^{N} \left( \log \left( 1 + \frac{\|f_j - f_\ell\|^2}{\frac{1}{N^2} \sum_{m,n} \|f_m - f_n\|^2} \right) - \log \left( 1 + \frac{\|z_j - z_\ell\|^2}{\frac{1}{N^2} \sum_{m,n} \|z_m - z_n\|^2} \right) \right)^2.$$
(4)

2. In the second step of minimization, the parameters of the prior-encoder ($\Psi_{PE}$) are trained for $k_2$ steps while parameters of the autoencoder ($\phi_E, \phi_D$) are fixed. The loss function for the prior-encoder consists of the NSW distance between the prior and posterior:

$$\mathcal{L}_{PE}(\mathbf{x}) = d_{NSW}((\Psi_E)_* P_X, (\Psi_{PE})_* \mu).$$
(5)

The pseudocode is outlined in Appendix B. Additional architecture and training details for the experiments in this paper are in Appendix C.

### 3.3 INTERPOLATION AND APPROXIMATE GEODESICS

EPSWAE attempts to learn a representation of the embedded manifold geometry that the data lies along. The question arises: how does one make use of this representation? A natural way of interpolating data which lies on a manifold is through geodesics or paths on the manifold which traverse dense regions. Here we present a network algorithm for efficiently approximating network-geodesics with relatively few, noisy samples from the prior, by encouraging connections through high density regions of latent space while minimizing total energy.

1. Gather samples of the posterior $(\Psi_E)_* P_X$, i.e., $\Psi_E(x)$ with $x$ is a minibatch of data. Additional prior samples can be used to supplement this if desired;

2. For each sample $j$, compute the *average* Euclidean distance $c_j$ of the k-nearest neighbors;

3. Generate a thresholded network: Let samples be represented as nodes. For a given threshold value $t$, sample $j$ is connected to sample $i$ with edge weight $d^h(i,j)$ if $d(i,j) < t \cdot c_j$. Here $h$ is the energy parameter in the edge weight (chosen to be 1 or 2 in experiments, with $h = 2$ encouraging shorter hops). Sample-specific thresholding, i.e., thresholding dependent on $c_j$, increases the probability of a 'central' node having a connection. Thus, this encourages (1) paths through high density regions, i.e., high-degree nodes, and (2) traversal of latent space through short hops as a consequence of localization of paths, i.e., paths existing only between nearby points due to thresholding.

4. Continue increasing $t$ until the graph is connected, i.e., there exists a path, direct or indirect, from every sample to every other sample. Then, use Dijkstra's algorithm (Dijkstra, 1959) to identify network-geodesics with least total energy through allowable paths on the thresholded network.

## 4 EXPERIMENTS

We run experiments using EPSWAE on three datasets. First, we use a 3D spiral data, where the latent space can be visualized easily, to demonstrate that the learned prior captures the geometry of the 3D spiral. We also present here the advantages of the network-geodesic interpolation over linear interpolation. Then we present interpolation and generation results on the MNIST (LeCun et al., 2010) and CelebA (Liu et al., 2018) datasets.

## 4.1 ARCHITECTURE

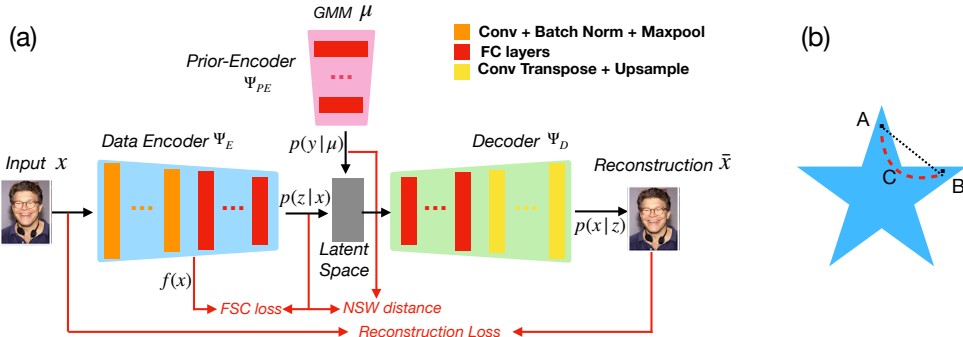

Figure 1: (a) is a schematic of the EPSWAE architecture. The red arrows indicate calculation of the loss terms. The prior-encoder generates a prior in latent space. (b) is simple example of a shape where interpolating between points A and B 'through the manifold' (red dashed line through point C) is desired, since linear interpolation (black line) leads out of the concave hull.

In all of the experiments, the prior-encoder $\Psi_{PE}$ consists of three fully connected hidden layers with ReLU activation. A sampleable distribution with dimension larger than the dimension of latent space is input to the prior-encoder which yields the prior. In principle, samples from any distribution can be the input $\mu$ to the prior-encoder. In the artificial data set we use a normal distribution, whereas for MNIST and CelebA we use a mixture of 10 Gaussians with random i.i.d means (from Gaussian with $\sigma = 2$). The schematic of the architecture is presented in Fig. 1(a). Note that the specifics of the layers shown in this schematic are for the image datasets MNIST and CelebA, the 3D Spiral data does not have convolutional layers. Details of the architecture, training, and hyperparameters for each dataset are given in Appendix C. The optimizer Adam (Kingma & Ba, 2014) with a learning rate of 0.001 was used for learning in both networks: the prior-encoder and the autoencoder.

## 4.2 LEARNING A LATENT MANIFOLD

We consider a 3D spiral randomly embedded to 40D space with noise as follows: (a) The formula for the spiral is given by $(x(t), y(t), z(t)) = (t \cos 7t\pi/4, t \sin 7t\pi/4, 2t)$, (b) a random $40 \times 3$ matrix is generated with i.i.d normal entries to map the spiral into $\mathbb{R}^{40}$, and (c) Gaussian noise with standard deviation 0.1 is added. The input to the prior-encoder is a Gaussian in $\mathbb{R}^{40}$, and the latent space is $\mathbb{R}^3$. For these tests, the data encoder, prior-encoder, and decoder all consist of three Fully Connected (FC) layers with 40 nodes each and ReLU activations.

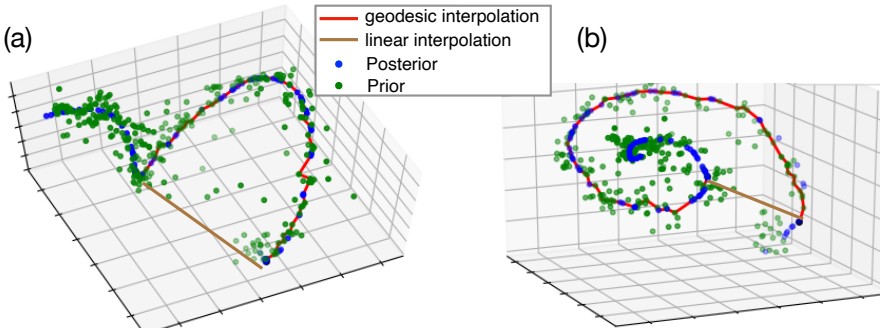

Figure 2: (a) side view and (b) top view of samples of the prior (green) and posterior (blue) in 3D latent space generated by EPSWAE after 100 epochs on the high dimensional input. The red line shows the interpolation along network-geodesics between two prior samples. The brown line shows the trajectory of linear interpolation between two points on the spiral.

As seen in Figure 2, after training, the posterior matches the spiral almost exactly and the prior learns the geometry of the spiral. A Nonlinear Sliced Wasserstein (NSW) distance and structural consistency term (where latent space is isometric with data space since convolutional layers aren't used) are used in the loss in Eqns. 3 and 5. A comparison highlighting the effects of these terms is shown in Fig. 5 in the Appendix D. We observe that the use of the NSW distance (as opposed to linear SW distance) improves the accuracy of the learned prior.

Manifold interpolation (in red in Fig. 2) uses the network-algorithm outlined in section 3.3. The interpolation is seen to have the desired form on the manifold, i.e., it approximates geodesics on the manifold. The larger the number of samples of the prior, the smoother the corresponding interpolation. In contrast, linear interpolation (brown line) between two points on the spiral does not capture the geometry at all, and goes through regions that are largely empty and untrained. In higher dimensional datasets, this may result in unrealistic interpolation. In order to traverse the manifold, both the prior-encoder (that allows the prior to capture the geometry of the spiral), and the network-interpolation algorithm are equally essential. Comparisons with baselines SWAE (Kolouri et al., 2018a) and VAE (Kingma & Welling, 2014) are shown in Appendix Fig. 6, and EPSWAE is seen to significantly outperform them in learning an improved latent representation.

**Computational cost**  The choice of nonlinearity was, in part, motivated by computational efficiency and simplicity, i.e., not requiring additional optimizations every evaluation or the training of a discriminator-like network to select optimal transformations. Our sinusoidal shear nonlinearity is closely related to a special case of the generalized SW distance defined in Kolouri et al. (2019) (see discussion in section 3.1 for differences ). Here, we present a comparison of computational time with two polynomial-type nonlinearities discussed in Kolouri et al. (2019) - cubic and quintic. Computations done on a laptop with Intel Core i7-10710 CPU, 16GB RAM, and computed over 1000 runs. In practice, the losses were indistinguishable with choice of nonlinearity (see Appendix E), whereas the sine shear is slightly less expensive (see Table 1). However, the complexity of cubic and quintic nonlinearities are exponential in dimension and hence are impractical for larger data.

Table 1: Comparison of computational time for nonlinearities in SW distance.

|  | **sine-shear NSW** | **cubic NSW** | **quintic NSW** |
| --- | --- | --- | --- |
| Computational Cost | **0.0050s $\pm$ 0.0006s** | $0.0054s \pm 0.0007s$ | $0.0060s \pm 0.0008s$ |

### 4.3 GENERATION AND OUTLIERS IN LATENT SPACE

In order to demonstrate one of the advantages of using a prior encoder in improving latent structure, we consider generation with an increased probability of sampling from outliers, i.e., by increasing the standard deviation of the distribution that feeds into the prior encoder. Studying generation from outliers may seem nonstandard, however, it is informative in the study of how data is encoded in the latent space. In Fig. 3 compare our results with the baseline SWAE (Kolouri et al., 2018a) in order to assess the effect of the prior encoder in improving latent structure. It is worth noting that while there exist several recent works that build upon SWAE for better generation, they do not attempt to solve the same problem as ours (learning latent manifolds). As seen in Fig. 3, the behavior of SWAE and EPSWAE at outliers is very different. For large $\sigma$, SWAE tends to generate unrealistic faces with distorted colorations, whereas EPSWAE is more likely to generate realistic faces (albiet with increased mode collapse at large $\sigma$). We see that EPSWAE encodes coherent information in a large region of latent space as a consequence of nonlinear transformations through the prior encoder. Since we are not competing with state of the art in generation here, we use downsized images and a small, unsophisticated model compared to state of the art methods.

Both SWAE and EPSWAE use the same data encoder and decoder, and are independently optimized: the encoder is (after downsizing CelebA images to $64x64$) Conv (3,16,3) $\rightarrow$ BatchNorm $\rightarrow$ ReLu $\rightarrow$ MaxPool (2,2) $\rightarrow$ Conv (16,32,3) $\rightarrow$ BatchNorm $\rightarrow$ ReLu $\rightarrow$ MaxPool (2,2) $\rightarrow$ Conv (32,64,3) $\rightarrow$ BatchNorm $\rightarrow$ ReLu followed by two FC layers of 512 and 256 nodes respectively with a leaky ReLu nonlinearity and the decoder is the reverse (replace Conv layers with Conv-transpose and MaxPool with Upsample). The prior-encoder consists of three FC layers from 186D $\rightarrow$ 128D latent space. See Appendix C for more details and hyperparameter values.

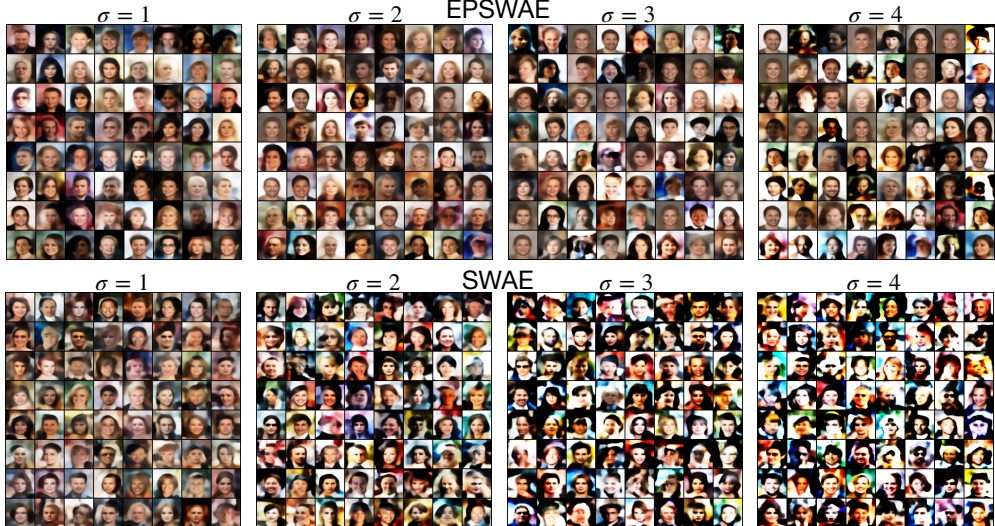

Figure 3: Images generated after 100 epochs from prior samples in (a) EPSWAE (b) Baseline SWAE at increasing standard deviations $\sigma$, i.e., progressively more 'outlying'.

We also present a comparison with SWAE on CelebA generation from samples around the mean in Appendix G, and find that EPSWAE sees a marginal improvement in the quality of faces. Generation on MNIST using EPSWAE (and comparisons with SWAE (Kolouri et al., 2018a) and VAE (Kingma & Welling, 2014)) are shown in Appendix F. MNIST results are obtained with $5D$ latent space. EPSWAE generation is more natural, whereas both other baselines generate some bloated and unidentifiable numbers.

Table 2: FID score comparison generated with 10000 samples (lower is better)

| SWAE | EPSWAE (bl) | EPSWAE (FSC only) | EPSWAE (NSW only) | ESPWAE (FSC + NSW) |
|------|-------------|-------------------|-------------------|--------------------|
| 178.04 | 161.55 | 162.14 | 161.06 | **157.86** |

For completeness, in Table 4.3 we present Frechet Inception Distance (FID) (Salimans et al., 2016) scores with 10000 images with $\sigma = 1$ (code from Seitzer (2020)). It is important to note that one can have good generation, despite learning a poor latent representation (and vice-versa), so scores like FID may not be the best way to evaluate the latent structure. However, comparison between SWAE and EPSWAE-baseline(bl) indicate that the prior-encoder significantly improves generation.

**Computational cost** : The computational cost of the different terms in the EPSWAE loss on the CelebA dataset is as follows: (1) FSC computation: $0.00066s \pm 0.0001s$, (2) linear SW distance computation: $0.01561s \pm 0.0005s$ (3) sine-shear NSW computation: $0.0150s \pm 0.0007s$ (with same number of total projections). All computations for CelebA were done on a workstation with one NVIDIA Tesla P100 GPU, two Intel Xeon E5-26660v4 CPUs, and 64Gb RAM.

## 4.4 INTERPOLATION

In this paper, we introduce a graph-based algorithm that interpolates over network-geodesics in curved latent manifolds, thus capturing properties of the geometry. Here we show that we obtain smooth interpolation using the network-geodesics algorithm, in contrast with linear interpolations that often tend to generate unrealistic intermediate images.

The start and end points correspond to real images (posterior samples). Figure 4(a) presents a comparison of linear interpolation (top) vs interpolation through network-geodesics (bottom) described

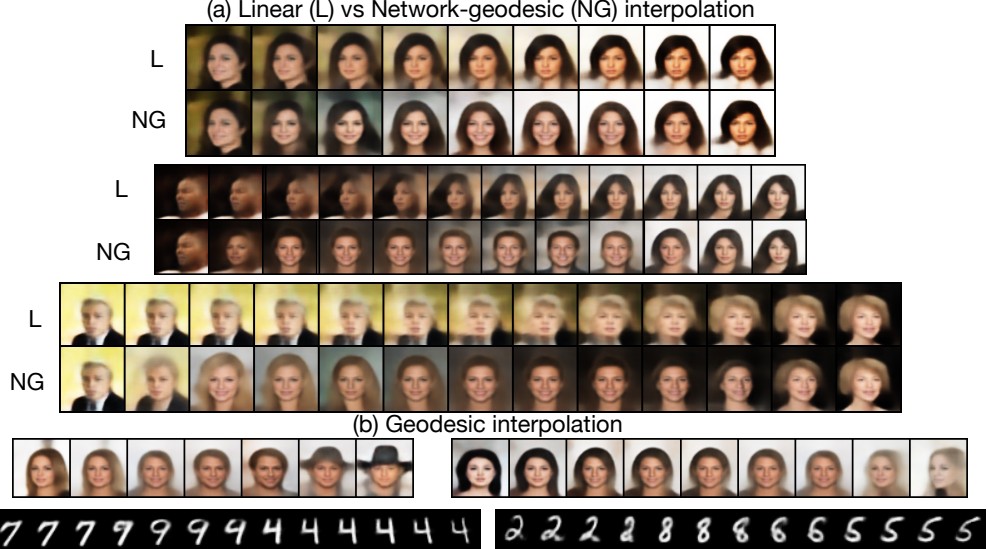

Figure 4: (a) Comparison of linear vs geodesic interpolation using EPSWAE. The top row of each row-pair is linear, and the bottom is geodesic-interpolation. (b) Randomly chosen instances for geodesic interpolations for MNIST and CelebA. The first and last images are reconstructions of real data, and the interpolations traverse through samples of the prior using the network-geodesic algorithm. A total of 400 samples are used in all cases.

in section 3.3 on CelebA. One can see that linear interpolations often go through unrealistic images (this corresponds to regions in latent space where training is limited), whereas geodesic interpolations go through more realistic and less blurry faces. This also serves as evidence, that the latent space for CelebA contains some natural structure that can be exploited.

Figure 4 (b) shows examples of interpolations on MNIST and CelebA datasets. MNIST interpolations are smooth and intuitive, for instance, in the bottom left MNIST panel in Fig. 4(b), the top part of a '7' first changes to a '9' naturally, followed by transformation of the '9' to a '4'. Interpolation along network-geodesics ensures that reconstructions of intermediate samples are realistic. Interpolations on CelebA are smooth and pass through intermediate images that could arguably pass for celebrities these days. The state of the art interpolations along network-geodesics on the manifold indicate that the learned prior does indeed encode the data manifold. For all interpolations shown in Fig. 4, we use energy parameter $h = 2$. Comparisons between interpolations corresponding to $h = 1$ and $h = 2$ are presented in Appendix 10. In experiments, energy parameter is not found to have a significant impact on the quality of interpolations. Additionally, comparisons of linear interpolation between equivalent networks of EPSWAE and SWAE are presented in Appendix I.

## 5 CONCLUSION

We introduce the Encoded Prior Sliced Wasserstein AutoEncoder (EPSWAE) that learns improved latent representations through training an encoded prior to approximate an embedding of the data manifold that preserves geometric and topological properties. The learning of an arbitrary shaped prior is facilitated by the use of the Sliced Wasserstein distance, which can be efficiently computed from samples only. We use a nonlinear variant of SW distance to capture differences between two distributions more efficiently, and employ a feature structural consistency term to improve the latent space representation. Finally, we introduce an energy-based algorithm to identify network-geodesics in latent space that maximize path density while minimizing total energy. We demonstrate EPSWAE's ability to learn the geometry and topology of a 3D spiral from a noisy 40D embedding. We also show that our model embeds information in large regions of latent space, leading to better outlier generation. Lastly, we show that our geodesic interpolation results on MNIST and CelebA are efficient and comparable to state of the art techniques. Our code is publicly available.

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

## A    BACKGROUND ON THE SLICED WASSERSTEIN DISTANCE

Wasserstein distances provide a natural metric for measuring distances between probability distributions based on optimal transport, i.e., the cost of deforming one probability distribution into another. For a measurable cost function $c(x, y)$ for $x, y \in \mathbb{R}^d$, and probability distributions $\mu$ and $\nu$ on $\mathbb{R}^d$, we define the $p$-Wasserstein distance between the distributions as

$$d_{W,p}(\mu, \nu) = \inf_{\Gamma \in \Pi(\mu,\nu)} \left( \int_{\mathbb{R}^d \times \mathbb{R}^d} c^p(x, y) \mathrm{d}\Gamma(x, y) \right)^{1/p}, \tag{6}$$

where $\Pi$ is the set of all joint distributions with marginals $\mu$ and $\nu$ (see Villani (2003)).

This Wasserstein distance between probability distributions is extremely difficult and computationally intensive in dimensions two and higher, i.e., $d \geq 2$. However, in dimension one, there exists a closed-form solution (Villani, 2003). A simple algorithm for computing the 1D Wasserstein distance is given as follows (see for example (Kolouri et al., 2018a)): (a) Generate $N$ one dimensional i.i.d samples $x_j \sim \mu$, $y_j \sim \nu$; (b) sort each list $X = [x_1, \ldots, x_N], Y = [y_1, \ldots, y_N]$ into ascending order denoted by $\tilde{X}, \tilde{Y}$ respectively; (c) define the approximation to the 1D p-Wasserstein distance

$$d_{W,p}(\mu, \nu) \approx \left( \frac{1}{N} \sum_{j=1}^{N} c^p(\tilde{x}_j, \tilde{y}_j) \right)^{1/p}, \tag{7}$$

where $\tilde{x}_j \in \tilde{X}, \tilde{y}_j \in \tilde{Y}$. The Sliced Wasserstein (SW) distance defines a metric on probability measures (Bonnotte, 2013) which provides an alternative to Eqn. 6 by exploiting the computational feasibility of the 1D Wasserstein distance in Eqn. 7. It involves averaging over one-dimensional orthogonal projections $\pi^\theta x := (\theta \cdot x)\theta$ as follows:

$$d_{SW}(\mu, \nu) = \left( \oint_{\mathbb{S}^{d-1}} d_{W,p}^p(\pi_*^\theta \mu, \pi_*^\theta \nu) dS(\theta) \right)^{1/p}, \tag{8}$$

The SW distance has seen a variety of implementations (Bonneel et al., 2015; Kolouri et al., 2018b; Lee et al., 2019).

The integral in Eqn. 8 can be easily approximated by sampling $M$ random one-dimensional vectors $\theta_k$ uniformly on $\mathbb{S}^{d-1}$ and computing

$$d_{SW}(\mu, \nu) \approx \left( \frac{1}{M} \sum_{k=1}^{M} d_{W,p}^p(\pi_*^{\theta_k} \mu, \pi_*^{\theta_k} \nu) \right)^{1/p}. \tag{9}$$

Several works (for example Nguyen et al. (2020); Deshpande et al. (2019)) have discussed reasons why this often does not result in the best computational method. Linear projections may be suboptimal for extracting information about the differences between $\mu$ and $\nu$, since a large number of linear projections may be required to get an accurate approximation for $d_{SW}$. Several works have suggested possible methods for improving the effectiveness of the SW distance (Chen et al., 2020b; Kolouri et al., 2019; Nguyen et al., 2020; Deshpande et al., 2019). In contrast, we use a Nonlinear Sliced Wasserstein (NSW) distance, an averaging procedure over (random) *nonlinear* transformations. For our goals, the choice of nonlinearity was motivated by the following considerations: (1) a bounded non-linearity would be beneficial since unbounded non-linearities (such as cubic polynomials) have a pronounced deformation on the tails of a measure and may excessively weight outliers. (2) A sigmoid is another potential candidate, but it saturates at high values and we want the non-linearity to be similarly effective everywhere. (3) The use of polynomials has been discussed in Kolouri et al. (2019), however, a full set of all higher order polynomials have exponential complexity and may be prohibitively expensive in high latent dimensions. We compared our method to cubic and quintic polynomials for the simple 3D spiral and found that the choice of the nonlinearity did not have a significant effect on performance, with our choice being the fastest (see section 4.2 and Appendix E for computational time and loss comparisons). In principle, other ensembles of nonlinear transformations could be used.

## B    Pseudocode

---

**Algorithm 1** Training EPSWAE

---

1:  **while** not converged **do**
2:      Update the autoencoder $\Psi_E, \Psi_D$:
3:      **for** $k_1$ substeps **do**
4:          Sample minibatch from data $\{\mathbf{x}^{(1)}, ..., \mathbf{x}^{(N)}\}$ with $\mathbf{x}^{(j)} \sim P_X$
5:          Compute feature extractor samples $\mathbf{f}^{(j)} = \Psi_{FE}(\mathbf{x}^{(j)})$
6:          Compute posterior samples $\mathbf{z}^{(j)} = \Psi_E(\mathbf{x}^{(j)})$
7:          Compute decoded output $\mathbf{x}^{(j)}_{recon} = \Psi_D((\mathbf{z})^j)$
8:          Generate samples $\{\xi^{(1)}, ...\xi^{(J)}\}$ with $\xi^{(j)} \sim \mu$
9:          Compute prior samples $\mathbf{y}^{(j)} = \Psi_{PE}(\xi^{(j)})$
10:         Compute reconstruction error: $\mathcal{L}_{recon} = \frac{1}{J} \sum_{j=1}^{J} d(\mathbf{x}^{(j)}, \mathbf{x}^{(j)}_{recon})$
11:         Compute NSW distance: $\mathcal{L}_{NSWdistance} = d_{NSW}(\frac{1}{N} \sum_{j=1}^{N} \delta_{\mathbf{z}^{(j)}}, \frac{1}{N} \sum_{j=1}^{N} \delta_{\mathbf{y}^{(j)}})$
12:         Compute FSC loss: $\mathcal{L}_{FSC} = d_{FSC}(\{\mathbf{f}^{(1)}, ..., \mathbf{f}^{(J)}\}, \{\mathbf{z}^{(1)}, ..., \mathbf{z}^{(J)}\})$
13:         Compute autoencoder loss $\mathcal{L}_{AE} = \alpha \mathcal{L}_{recon} + \beta \mathcal{L}_{NSWdistance} + \kappa \mathcal{L}_{FSC}$
14:         Compute gradients of $\mathcal{L}_{AE}$ wrt to $\phi_E, \phi_D$
15:         Update $\phi_E, \phi_D$
16:     **end for**
17:     Update the prior-encoder $\Psi_{PE}$:
18:     **for** $k_2$ substeps **do**
19:         Sample minibatch from data $\{\mathbf{x}^{(1)}, ..., \mathbf{x}^{(J)}\}$ with $\mathbf{x}^{(j)} \sim P_X$
20:         Compute posterior samples $\mathbf{z}^{(j)} = \Psi_E(\mathbf{x}^{(j)})$
21:         Generate samples $\{\xi^{(1)}, ...\xi^{(J)}\}$ with $\xi^{(j)} \sim \mu$
22:         Compute prior samples $\mathbf{y}^{(j)} = \Psi_{PE}(\xi^{(j)})$
23:         Compute prior-encoder loss: $\mathcal{L}_{PE} = d_{NSW}(\frac{1}{N} \sum_{j=1}^{N} \delta_{\mathbf{z}^{(j)}}, \frac{1}{N} \sum_{j=1}^{N} \delta_{\mathbf{y}^{(j)}})$
24:         Compute gradients of $\mathcal{L}_{PE}$ wrt to $\phi_{PE}$
25:         Update $\phi_{PE}$
26:     **end for**
27: **end while**

---

## C    Architecture and Training Details

For all datasets, the prior-encoder $\Psi_{PE}$ consists of three fully connected hidden layers and ReLU activations. For all datasets, the autoencoder and prior-encoder losses (given in Eqns 3 and 5) are trained iteratively using the optimizer Adam (Kingma & Welling, 2014) with a learning rate of 0.001. We experimented with both $p = 1$ and $p = 2$ (corresponding to p-Wasserstein) in the SW distance and did not find any significant differences; all results in this paper use $p = 2$. For each calculation of the NSW distance, $L = 5$ random nonlinear transformations were taken followed by $M = 50$ one dimensional projections per transformation. Data-specific model and parameter details are given below.

**3D Spiral dataset**: The input to the prior-encoder is a 40D Gaussian, and the latent space is 3D. The input to the autoencoder is a 40D embedding of a 3D spiral manifold with 10% noise. The dataset consists of 10000 samples, and a batch of 100 was used. The prior-encoder, the data encoder, and the decoder consist of three Fully Connected (FC) layers with 40 nodes each and ReLU activations. The reconstruction loss is given by the Mean Square Error and $\alpha = 1, \beta = 0.1, \kappa = 0.01$ in Eqn 3. In the absence of convolutional layers, the FSC term encourages the pairwise distances of the minibatch in latent space to be similar to the pairwise distances of the minibatch in the data space. The prior-encoder is trained $k_1 = 2$ times for each training of the autoencoder $k_1 = 1$. Power of distance $h = 2$ is used to compute edge weights for computing network-geodesics.

**MNIST dataset**: The input to the prior-encoder is a 40 dimensional *mixture of 10 Gaussians*, and the latent space is 5 dimensional. The data encoder takes MNIST images as input using a batch size of 100, and consists of the following layers Conv $(1,10,3) \rightarrow$ BatchNorm $\rightarrow$ ReLu $\rightarrow$ MaxPool

$(2,2) \rightarrow$ Conv (10,16,3) $\rightarrow$ BatchNorm $\rightarrow$ ReLu followed by two FC layers of 512 and 256 nodes respectively with a leaky ReLu nonlinearity. The encoder outputs a 5 dimensional latent representation. The decoder consists of the reverse, i.e., three fully connected layers of size 256, 512, and 1936 nodes respectively. This is followed by ConvTranspose (16,10,3) $\rightarrow$ LeakyReLu $\rightarrow$ Upsample(2,2) $\rightarrow$ ConvTranspose (10,1,3) $\rightarrow$ Sigmoid. The decoder output has size $28 \times 28$. The prior-encoder is trained $k_1 = 1$ times for each training of the autoencoder $k_2 = 1$. The reconstruction loss is given by the Binary Cross Entropy and $\alpha = 1, \beta = 0.1, \kappa = 0.001$ in Eqn. 3. The FSC loss encourages the pairwise distances of the minibatch in latent space to be similar to the pairwise distances of the minibatch in *feature* space computed at the output of the last convolutional layer in the data encoder. Energy parameter $h = 2$ is used to compute edge weights for computing network-geodesics.

**CelebA dataset**: The input to the prior-encoder is a 186 dimensional *mixture of 10 Gaussians*, and the latent space is 128 dimensional. We ran experiments with different latent dimensions and found that the generation ability didn't vary significantly as a function of latent dimension. We trade-off image quality of the input images for computational speed by downsizing the input and employ a fairly simple network compared to state of the art computer vision architectures that use CelebA. The data encoder takes CelebA images (of size $218 \times 178 \times 3$) and downsizes them to size $64 \times 64 \times 3$. The data encoder consists of the following layers Conv (3,16,3) $\rightarrow$ BatchNorm $\rightarrow$ ReLu $\rightarrow$ MaxPool (2,2) $\rightarrow$ Conv (16,32,3) $\rightarrow$ BatchNorm $\rightarrow$ ReLu $\rightarrow$ MaxPool (2,2) $\rightarrow$ Conv (32,64,3) $\rightarrow$ BatchNorm $\rightarrow$ ReLu followed by two FC layers of 512 and 256 nodes respectively with a leaky ReLu nonlinearity. The encoder outputs a 128 dimensional latent representation. As in the case of MNIST, the decoder consists of the reverse, with convolutions replaced by Convolution Transpose, and MaxPool replaced by Upsample. The output of the decoder is passed through a sigmoid nonlinearity and is of size $64 \times 64 \times 3$. The prior-encoder was trained $k_1 = 1$ times for each training of the autoencoder. The reconstruction loss is given by the Binary Cross Entropy and $\alpha = 500, \beta = 50, \kappa = 0.05$ in Eqn. 3. The FSC loss encourages the pairwise distances of the minibatch in latent space to be similar to the pairwise distances of the minibatch in *feature* space, i.e., computed at the output of the last convolutional layer in the encoder. Energy parameter $h = 2$ is used to compute edge weights for computing network-geodesics.

## D    SPIRAL BASELINE COMPARISONS

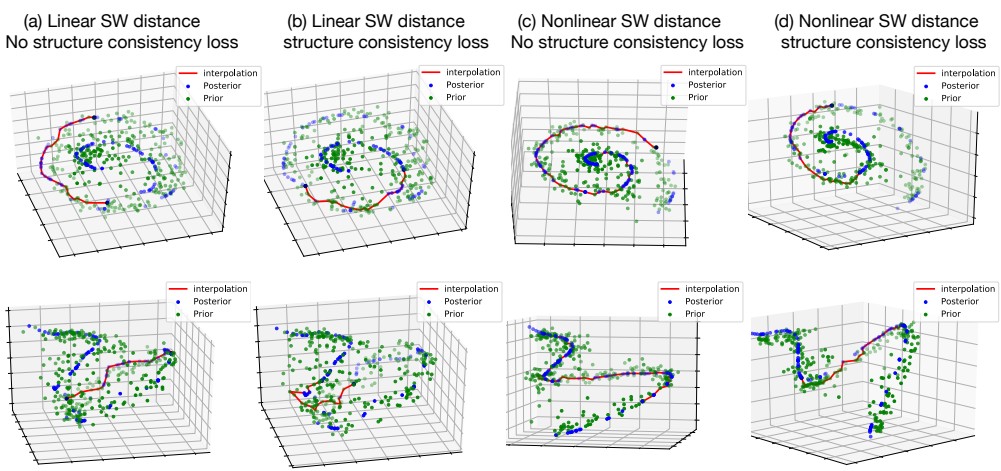

Figure 5: Comparisons of the EPSWAE model with different loss terms. Top panels present top views, and bottom panels present the corresponding side views. The red curves show interpolation between two randomly selected samples using the network-geodesic algorithm. All figures are generated after 100 epochs with a lr=0.01, and batch size =100. $k_1 = 1, k_2 = 2$.

Figure 5 shows the effects of the different loss terms in EPSWAE on the geometry of the learned prior and posterior. The position and orientation of the spiral in the 3D plots are random and the views

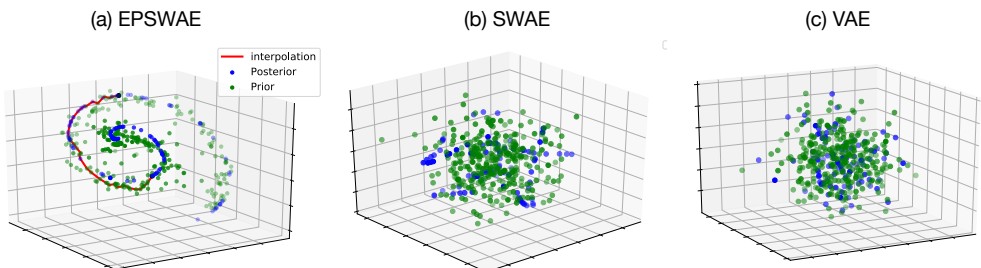

Figure 6: Comparison of EPSWAE with baselines SWAE (Kolouri et al., 2018a) and VAE (Kingma & Welling, 2014). All figures are generated after 100 epochs with a lr=0.01, and batch size =100. For EPSWAE, $k_1 = 1, k_2 = 2, \alpha = 1, \beta = 0.1, \kappa = 0.001$.

in the image are hand-chosen to be equivalent. (a) shows latent space and interpolations (red) using EPSWAE with a linear Sliced Wasserstein distance in the loss, and no structural consistency term. (b) shows that adding a structural consistency term doesn't remarkably improve the quality of the manifold learned, however, consistent with other experiments, it seems to improves interpolation slightly. Note here that since we don't use convolutional layers for the 3D spiral, the stuctural consistency term preserves distance in latent space corresponding to distances in data space. (c) shows that employing the NSW distance term significantly improves the learned structure in latent space. The improvements resulting from incorporation of the NSW and structural consistency terms as seen in these visualizations of the 3D spiral lead us to use both loss terms (as in (d)) on all results in the main paper.

Figure 6 compares the EPSWAE model with baselines VAE and SWAE. The VAE uses KL divergence in the loss, which constrains the prior to be sampled from a Gaussian distribution. While the vanilla SWAE uses the SW distance, the prior remains a Gaussian, leading to an unnatural embedding of the data manifold. In contrast, EPSWAE results in a significantly better learned prior as a consequence of the nonlinear SW distance and the prior-encoder network trained explicitly to improve the latent representation.

## E    COMPARISON WITH OTHER SLICED WASSERSTEIN NONLINEARITIES

Here, we compare our choice of nonlinearity in the SW loss (sinusoidal shear) with some other common nonlinearities. While there exist several methods to improve the SW distance (e.g. max-SW (Deshpande et al., 2019) which has seen several subsequent variations), these involve additional training, and hence aren't considered (although many such versions of SW distance could be used in conjunction with out method). Instead we consider the polynomial generalized Radon transforms described in (Kolouri et al., 2019). The computational cost of a single polynomial generalized Radon transform is $\mathcal{O}(d^k)$ for dimension $d$ and $k^{th}$ order of polynomial, and hence we limit our investigation to cubic and quintic polynomials. We show results on the artificial 3D Spiral dataset trained upto 100 epochs with $\alpha = 1$, $\beta = 0.1$, $\kappa = 0.01$. We see in Fig. 7 that the choice of nonlinearity does not have a significant effect on the loss, however, as seen in Table4.2, the sine-shear has slightly lower computational cost.

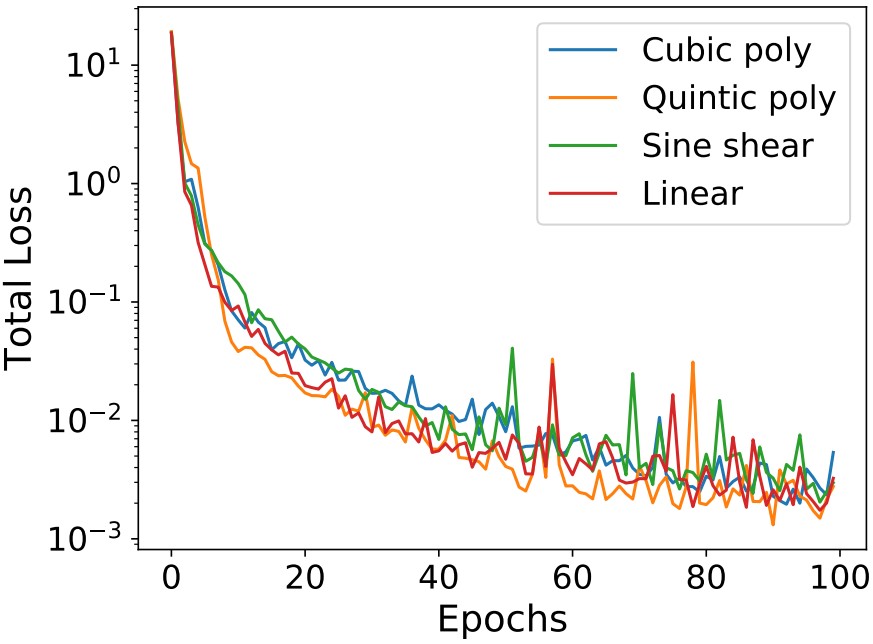

Figure 7: Loss as a function of training epochs for linear SW, cubic nonlinearity, quintic nonlinearity and our (sine-shear) NSW distance. A full set of cubic and quintic functions terms are considered for computing the generalized Radon transform in the SW distance. $L = 5$ nonlinear transforms with $M = 50$ linear 1D projections each taken for all cases.

## F  MNIST GENERATION RESULTS

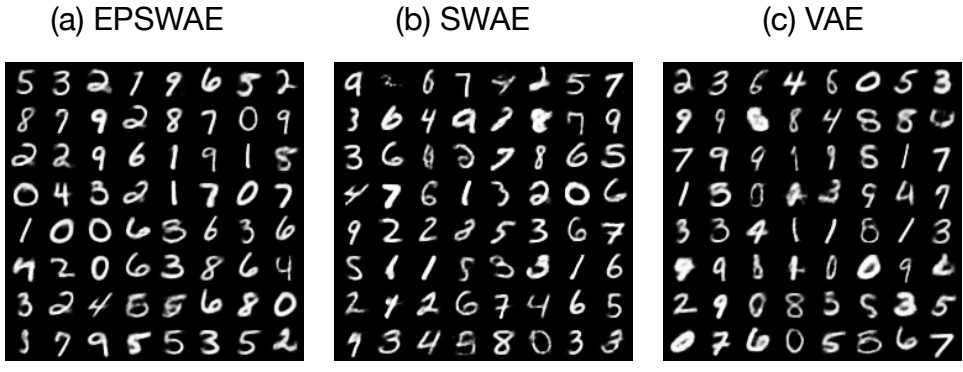

Figure 8: Comparison between (a) EPSWAE (b) SWAE and (c) VAE for generation on MNIST. for all three, batch size = 200, epochs=100, lr = 0.001, for EPSWAE $k_1 = 2$, $k_1 = 1$, $\alpha = 1$, $\beta = 0.1$, $\kappa = 0.001$.

Figures 8 (a,b,c) shows generation on the MNIST dataset after 100 epochs on EPSWAE, baseline SWAE(Kolouri et al., 2018a), and baseline VAE (Kingma & Welling, 2014) respectively. All networks EPSWAE and baselines SWAE and VAE use equivalent architectures (outlined in section of Appendix C), and hyperparameters are optimized individually. As seen in the figure, we observe that EPSWAE generated samples were consistently found to have a lower fraction of 'false' digits, i.e., digits that are unrealistic.

## G    CELEBA GENERATION RESULTS

(a) EPSWAE                                    (b) SWAE

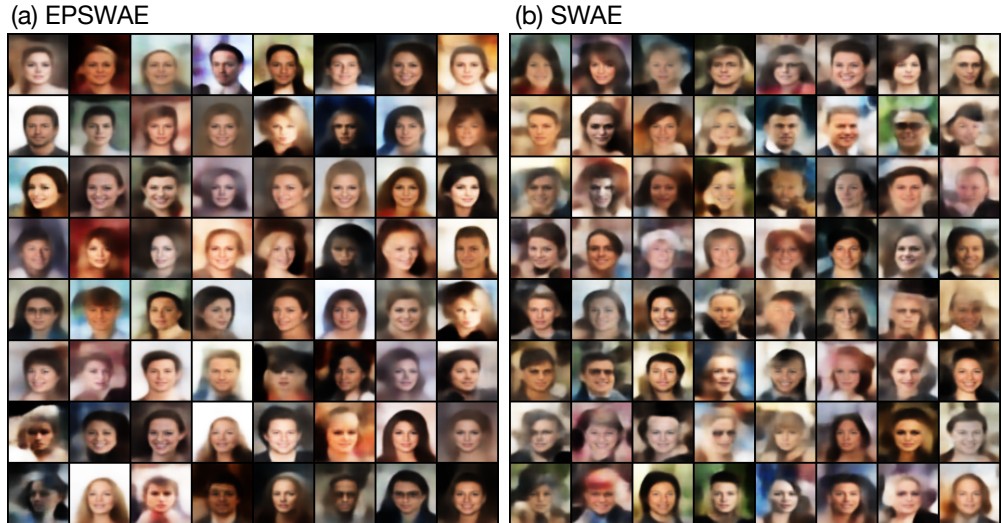

Figure 9:   Images generated from prior samples in (a) EPSWAE (b) Baseline SWAE. Batch size = 200, epochs=100, lr = 0.001. $\alpha = 500, \beta = 50, \kappa = 0.05$.

CelebA raw images are downsized to $64 \times 64$ pixels, and a simple architecture is employed (see Appendix C). This serves as a proof of principle for a better latent representation and interpolation, without a high computational cost. Naturally, using a more sophisticated network (for instance ResNet, VGG etc.) without downsizing the data would yield higher quality images at a computational cost.

Figure 9 shows generation on the CelebA dataset after 100 epochs with (a) EPSWAE and (b) baseline SWAE (right) respectively. Both employ equivalent architectures and take downsized images as input. As seen in the figure, EPSWAE generated images are more realistic. Note that this comparison is for equivalent training epochs and architectural details, and does not make claims about the quality of SWAE images with longer training.

## H    EFFECT OF ENERGY PARAMETER ON CELEBA INTERPOLATIONS

Additional CelebA interpolations are shown in Fig. 10. The length of interpolations is automatically selected by the network algorithm. In some cases, a linearly interpolated point is added between every two samples on the network-geodesic to make smoother transitions; this would be unnecessary with enough samples of the prior, however it could be impractical in high latent dimension to generated sufficiently many prior samples. Note that even if only one intermediate point is selected by the network geodesic, the generated interpolation could be significantly different from a purely linear interpolation (an illustration is provided in Figure 4.1 - the direct linear interpolation from A to B is significantly different from the interpolation through the point C). Fig. 10 presents a comparison between energy parameter $h = 1$ and $h = 2$. The energy parameter determines the power of the distance metric used to compute the edge weight. One can think of the higher value ($h = 2$) as corresponding to stronger connections between points, and encouraging shorter hops. In practice, as seen from the figure, there is no clear advantage to choosing a specific value of energy parameter $h$.

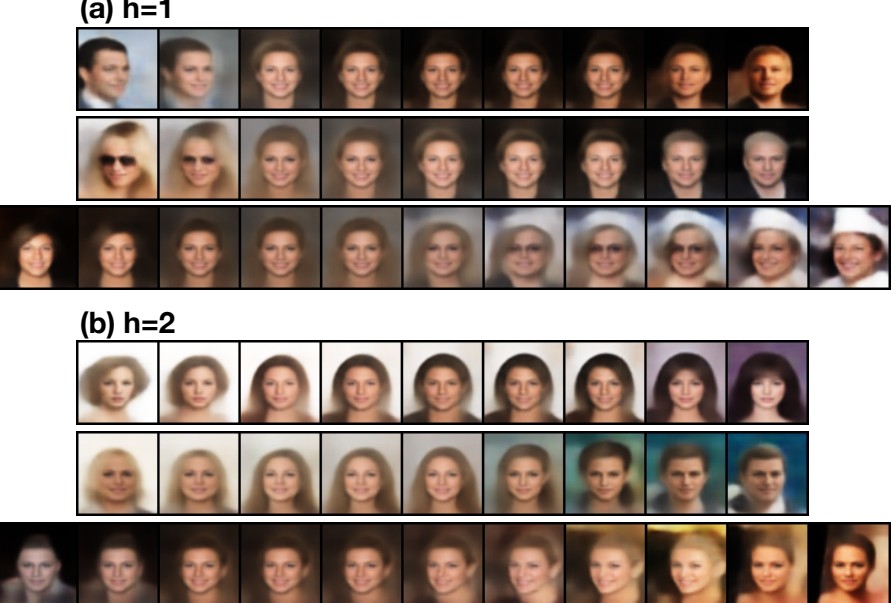

Figure 10: Additional CelebA interpolations. The first and last images are reconstructions of real data, and the interpolations traverse through samples of the prior using the network-geodesic algorithm. (a) Top three panels show interpolations for energy parameter $h = 1$, and (b) bottom three panels show interpolations for energy parameter $h = 2$. Latent space is 128 dimensional. Hyperparameters are the same as those used in the paper. A total of a 400 samples in latent space are used.

# I   LINEAR INTERPOLATION COMPARISON EPSWAE AND SWAE

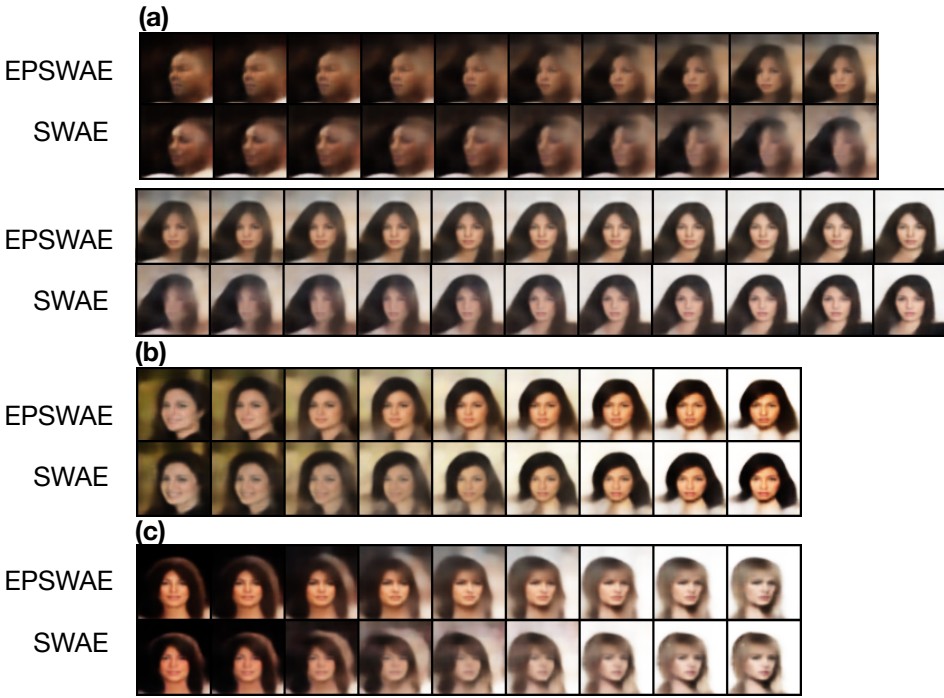

Figure 11: Comparisons of linear interpolation between EPSWAE and SWAE. The first and last images are reconstructions of randomly picked real data (a) shows long interpolation (over two lines), (b,c) show shorter interpolations. Hyperparameters are the same as those used in the paper. Latent space is 128 dimensional. A total of a 400 samples in latent space are used.

We compare here linear interpolations on the CelebA dataset for MNIST and CelebA. The autoencoder networks used are identical and outlined in section C. In contrast, EPSWAE uses the additional prior encoder. Both models are independently optimized, and the interpolations are linear. As seen in Fig. 11, interpolations in EPSWAE are more realistically identifiable as 'faces' than SWAE, which mixes up the features and generates blurry intermediate images. This is suggestive that the prior-encoder may play a role in improving latent representation.

