# OpenReview forum: "Encoded Prior Sliced Wasserstein AutoEncoder for learning latent manifold representations"
_ICLR.cc/2021/Conference — Reject_

### Official Review · AnonReviewer4 · 2020-10-27
**This paper addresses the representation learning of VAEs and is reasonable, but need to dig more.**

**Rating:** 5
**Confidence:** 4

**Review:**

This paper addresses the issues of representation learning with VAEs and propose EPSWAE as a solution. EPSWAE applies a prior encoder to construct an implicit prior, which is more flexible. Moreover, the authors apply the sliced Wasserstein distance for the matching between the posterior and the prior, enhance the conventional SWD with non-linear transformations and make the latent space similar to the feature space with a structural consistency loss. This paper also proposes a graph-based algorithm for minimizing the pathwise energy to achieve the manifold walking to improve the interpolation in the latent space.

-------------------

The paper is well written. The pipeline is clear and easy to understand. The representation learning with VAEs is a widely studied topic. Using a flexible implicit prior will boost the learning of the latent codes. The usage of the encoder (or generator in the adversarial cases) in the latent space is widely discussed in previous works, such as vampprior, semi-implicit VI, doubly semi-implicit VI, etc. Thus the contribution in this part is limited.

The sliced Wasserstein distance is an efficient approximation of the Wasserstein distance for the distribution matching. To avoid the projections that contain useless information, as cited in this paper, a lot of papers generalize the typical random linear transform to non-linear transformation (Chen et al., 2020b; Kolouri et al., 2019; Nguyen et al., 2020; Deshpande et al., 2019). Here the authors propose the non-linear sliced Wasserstein (NSW) distance in Equation 5 with a transformation shown in Equation 6, which appears to be a special case that satisfies the four conditions discussed in (Kolouri et al., 2019). Is there any difference between NSW distance and the generalized SWD in Kolouri et al., 2019?

The graph-based method is an interesting way for the manifold walking and is much better than conventional ways such as linear interpolation. I think this part should be discussed more in the paper.

-------------------

Some detailed questions about the technique

* The authors claim the usage of FSC encourages pairwise distances of the latent code to be similar to the pairwise distances of the data features. I am curious if FSC is necessary for manifold learning. In figure 5, it does not show much difference with/without the FSC loss.

* As claimed in the paper, the adversarial methods in latent space are expensive, while in the experiment part, there is no computation (such as time per update step) comparison with the adversarial methods. The usage of FSC also needs to compute the pairwise distance. Is that expensive too?

* In the abstract and the pseudocode shown in the appendix, the prior encoder is trained with sliced Wasserstein loss. Why we choose SWD for the training instead of NSW?

* It is also confusing that in the method part, the prior encoder is trained with NSW loss, which is not consistent as claimed in the other parts of the paper.

* In the experiments, the authors claim the generations are better, while the quantitative results, such as fid, are not provided.

* For the spiral toy dataset: views in Fig. 5 seem not consistent and are hard to compare. How about we compare in 2d cases?

In conclusion, the paper has the merits, and these investigations may be helpful for this problem, but it is not enough and need to dig out more to be an ICLR publication.

---

Update after the discussions

I appreciate the efforts that the authors make in their responses, some of which address my concerns and improve the quality of the paper.

I have raised my rating. However, taking into account all information during the discussion phase, I stick to my original review that this paper still needs to explore more to be a mature publication. For example, if the main contribution comes from the prior encoder, as I said the contribution is limited since the usage of the encoder (or generator in the adversarial cases) in the latent space is widely discussed in previous works, such as vampprior, semi-implicit VI, doubly semi-implicit VI, etc. This also seems to make the contribution of the sliced Wasserstein part incremental. Plus, as this paper has several components, their relations need to be discussed in a more clear way. Thus, more ablation studies are needed to help this paper to present its insight in a much more clear way.

Thanks again for the efforts that the authors make and I hope my reviews could help them to polish this paper to be a nice publication.

---

> ### Author Response · Authors · 2020-11-18
> **Expanded discussion on NSW distance, elucidate goals more clearly, provide timing tests and quantitative comparisons**
>
> We thank the referee for their positive assessment of the
> importance of this work, and our contributions. We have made several changes in response to the referees comments, and think the resulting changes have clarified a lot of details, improved our results (based on feedback from the referees), and strengthened our manuscript. We hope you will take the take the time to evaluate this version.
>
> The notable changes are:
> (a) better communication that the goal is improved latent representation and encoding the data manifold (and not generation as was probably miscommunicated), (b) including comparions of outliers generated from EPSWAE and baseline SWAE where EPSWAE far outperforms SWAE demonstrating learning in a larger region of latent space, (c) comparison of linear vs geodesic interpolation on CelebA, which shows more realistic intermediate images through geodesic interpolation (d) inclusion of quantitaive FID scores and computational times (including comparison of computational times with other nonlinearities in literature that show that our nonlinearity is simple and cheap). (e) rewording to emphasize our motivation for chosing the specific nonlinearity - it was brought to our attention that the nonlinearity is closely related to the generalized sliced wasserstein distance in Kolouri et al 2019 (we clarify this, cite a bunch of related literature, present a comparison between several equivalent nonlinearities, and discuss the motivation for choosing the specific form of nonlinearity relating to boundary considerations).
> Several smaller changes, and additional figures have been included in the Appendix.
>
> Here we provide a point by point response (with R denoting referee comment and A denoting our response):
>
> R: Is there any difference between NSW distance and the generalized
> SWD in Kolouri et al., 2019?
> A:  We have included a detailed discussion on the motivation behind our choice of nonlinearity, and comparisons with others in literature in the intro, background, section 3.1 and appendix A. The gist is as follows:
> In fact, it has been brought to our attention that the sine-basednon-linearity is almost an example of a generalised Radon transform discussed in Kolouri et. al. 2019 which discusses polynomial non-linearities. The two concepts are not quite the same (for example, our nonlinear transformations do not satisfy their condition H2). We chose our nonlinearity with the following motivations. A bounded non-linearity would be beneficial as unbounded non-linearity (such as cubic polynomials) have a pronounced deformation on the tails of a measure and may excessively weight outliers, which may be undesirable.
>
> The cubic (or higher order) nonlinearities also require O(L^3) flops where L is the number of latent dimensions, which is much more expensive, and not scalable to higher dimensional datasets. A sigmoid is another potential candidate, but it saturates at high values and we want the non-linearity to have an effect everywhere. We also wanted a way to easily adapt the nonlinearity to the distribution in a straightforward and efficient way (done by choosing the frequency that are most likely to produce deformations which highlight differences in the measures) - this breaks condition H2 in Kolouri et. al. 2019. A sine was the natural choice from a mathematical view-point.
> However, in practice, we do not expect a significant difference between reasonable, sufficiently rich choices of nonlinearity. Kolouri et. al. 2019 offers a rich discussion on the closely related generalised SWD are useful and compares several possibilities and against max-SWD approaches. We have done some studies on our spiral data set and found
> that the cubic and quintic GSWD discussed in Kolouri et al have essentially the same effect as the sine (although do not generalise as easily to higher dimensional data sets).
>
> We have restructured the intro, background, and sections 3.1 and appendix A (both relating to nonlinear SW distances) in our paper to do the following: (1) provide the above theoretical motivation for the choice of nonlinearity; (2) provide a more clear discussion about the existing work regarding improvements to SWD, such as the GSWD; (3) clearly state that using a nonlinear transformation is not itself a novel contribution; (4) provide comparisons with the cubic and quintic GSWD discussed in Kolouri et al. 2019

---

> > ### Author Response · Authors · 2020-11-18
> > **Point by point response**
> >
> > R: The graph-based method is an interesting way for the manifold walking and is much better than conventional ways
> > A: Thanks for the positive evaluation! We agree that this is an interesting approach to manifold walking, and we have taken your
> > advice to expand the discussion on the interpolation, which highlights better the main contributions of our paper. We have also included additional interpolations and comparison with straight-line linear interpolations in the appendix on CelebA (Fig. 4 and Appendix I).
> >
> > Now, to the detailed questions:
> >
> > 1. Sainberg et al `18 presents a good discussion on why any type of structural consistency is useful. We are interested in
> > extracting underlying geometrical properties of the data. The abstract features (as opposed to the direct input), one would hope, if the encoder is doing its job, may already be capturing some aspects of the underlying data structure, so a structural consistency wrt them that encourages feature-isometry should naturally help with encoding structural properties of the data (although I imagine this could a more important role in video data etc with inherent temporal structure). Note that improved generation is not necessarity a measure of improved latent embedding (which makes the precise contribution of the FSC term difficult to measure quantitavely).
> >
> > 2. This is a good point. We have added a cost analysis at the
> > end of section 4.3. In our experiments, computing the FSC term is very quick:it takes about half the time required to compute the sliced Wasserstein distance and moreover, the vanilla linear SWD and the nonlinear SWD with sine nonlinearity take the same amount of time to within error margins (this is ridiculously fast, and one of the reasons we picked this nonlinearity over several others that require additional training). We have also added a cost comparison of some polynomial-nonlinearities (discussed in Kolouri 2019) that don't require additional network optimization in Table 1 :)
> >
> > As you may imagine, a computation of simple pairwise distances, is significantly faster than adversarial training for two reasons: (1) fairly simple distances are calculated between low-dimensional features (at the output of the convolutional layers). (2) this is done within a batch (for reasonable batch sizes, the computational cost is low).
> >
> >
> > 3. This is entirely a typo. We do use NSW distance for training. Thanks for pointing  it out.
> >
> > 4. This is also part of the same typo as before, which may have been confusing. We still use NSW distance. We have corrected this
> >
> > 5. We have included FID scores in Table 2, including not only SWAE vs EPSWAE but also the FID scores
> > involving the same model and hyperparameters trained (A) with standard SWD and no FSC term; (B) with nonlinear SWD and no FSC term; (C) with standard SWD and FSC term; and (D) both nonlinear SWD and FSC term. The latter has the lowest FID score. There is a big jump between any of the EPSWAE variants and SWAE. These results suggest that our improved latent representation is leading to improved generation according to the FID score. We should emphasize again that our models are small and unsophisticated compared with state of the art generative models. We have tried to re-write the paper to better clarify the strengths of the method we are using. Note that it is possible to have good generation despite learning a poor latent space, so scores like FID may not be the best way to evaluate the advantages of the method.
> >
> >
> > 5. We apologize for the confusion. The spirals are not fixed in 3D space, but learned embeddings of the data (a 3D spiral embedded in 40D by a random linear transformation with noise) by the encoder and prior-encoder. Hence, the embedding is not oriented the same way each time, but we have tried to match the perspective visually. 3D spirals make a clearer point that the prior learns the geometry of the data (which is significantly harder in 3D than in 2D)!

---

> > > ### Comment · AnonReviewer4 · 2020-11-23
> > > **A few more discussions about the paper**
> > >
> > > I appreciate the authors carefully address my questions and make the revisions in the updated paper. The paper is much more clear than the previous version to me, and I am happy to see the authors discuss more in the difference between NSW and GSWD, as well as the difference between linear interpolation geodesic interpolation.
> > >
> > >
> > > I agree that making the projected logits bounded can make the representation more robust. The sinusoid projection also lowers the computation complexity, which is a good point against polynomial projections. Table 1 has shown support for the second point. However,   one concern is that Fig. 7 does not show (or not clearly show) the difference between Sine shear and the other projections in the perspective of robustness. I guess the loss in training is not a good metric to support the first point. One possible way is to compare the FID of EPSWAE with cubic NSW and with sinusoid NSW, though this is not a direct way to compare the effect of the transformations on representations.
> > >
> > >
> > >
> > > Another curiosity is how important is the prior encoding to the model. If the prior encoder is not trained ($k_2=0$), what will happen to the representations?
> > >
> > >
> > >
> > > To my understanding, in this paper, three components, i.e., NSW, FSC, prior-encoding, jointly makes the representation stronger and allows the geodesic interpolation. A question is which part plays a more important role in learning a better manifold representation. This is more important, and it is possible to provide more insights into other works. I am happy to see the authors' opinion and discuss with the authors.

---

> > > > ### Author Response · Authors · 2020-11-23
> > > > **Addressing the comments. the prior encoder is the most important component for capturing data structure (FSC and nonlinearity are additional helpful regularizations).**
> > > >
> > > > Thank you for your positive evaluation and acknowledging the improvements to the paper as a result of addressing referee comments.
> > > >
> > > > We are glad to see that our arguments for choice of nonlinearity and why it’s faster and expected to be more robust are supported by the referee. In practice, polynomial nonlinearities scale exponentially with latent dimension (d), which makes them prohibitive. Given a latent dimension d, using a full set of cubic nonlinear polynomials (as in Kolouri 2019) has computational complexity O(d^3), and for quintic is O(d^5) per nonlinear projection. In contrast our sine-shear has the same O(d) complexity per nonlinear projection.
> > > > As you can see for our (relatively simple) network with d=128 latent space for CelebA, even cubic (let alone other higher order polynomials) is prohibitive, for no added advantage. This makes our nonlinearity much more desirable for image data (with generally high dimensionality). We state this in the section discussing nonlinearity.
> > > >
> > > > We answer the next two questions together below:
> > > >
> > > > We view the paper as having two primary contributions:
> > > > (1) learning better representation to capture nontrivial nonconvex data manifolds (as seen in Fig. 2). This simply cannot be done for complex date manifolds with an unlearned or constrained prior, hence the prior encoder is required here. Without learning the prior encoder (k2=0), the model is simply SWAE with a fixed prior- whether the fixed prior is gaussian or a deformed gaussian makes no difference for an arbitrary dataset (the SWAE paper studies this and finds no significant improvements for different fixed priors). Fig. 6 of our paper shows that SWAE simply isn’t able to learn the geometry of the 3D spiral, whereas EPSWAE does this quite well, primarily thanks to the prior encoder.
> > > >
> > > > (2) The second major contribution is the network-based interpolation technique to interpolate along natural manifold geodesics. With our method, typically larger regions in latent space (wrt prior measure rather than Lebesgue measure) will see training (Figure 3 speaks to this aspect), and interpolations will go through fewer untrained regions/holes (Figure 4 speaks to this aspect) since natural manifolds are locally smooth.
> > > > Both the FSC term and the nonlinearity are additional regularisations that help with efficient encoding of the manifold - they are not expected to have a massive effect, but do encourage improved latent representations for reasons discussed in the paper  (ablation studies Fig.5 and FID scores support this. Several other papers also support this claim, and we cite these papers). Thus, our paper suggests that if one wants to learn the geometry/structure of the manifold, the prior encoder is the most important component.
> > > >
> > > > We’d like to sincerely thank you for your feedback and comments and are happy to discuss further. We believe our work presents a theoretically sound solution to a very important problem in the field. We have addressed all the referees concerns and hope that our much improved discussion and results make it acceptable for publication. We'd appreciate it if you could update your score to reflect your assessment of our updated version. Thank you!

---

### Official Review · AnonReviewer1 · 2020-10-28
**missing some related works, unclear motivation, and insufficient evaluation**

**Rating:** 5
**Confidence:** 4

**Review:**

The paper introduces an additional prior-encoder network to autoencoders to learn an unconstrained prior. The autoencoder and prior-encoder networks are iteratively trained with the sliced Wasserstein distance (SWD). To strengthen SWD, this paper further applies nonlinear transformations with a structural consistency term for better match between two distributions. For better interpolation on the latent space, it also introduces a graph-based algorithm.

While the paper cites some works that also aims at addressing the drawback of SWD, it still misses some important related works like [a, b, c]. The paper is highly expected to make more discussion and suitable empirical comparison with them.

[a] Deshpande et al., Generative modeling using the sliced wasserstein distance, CVPR 2018.

[b] Wu et al., Sliced wasserstein generative models, CVPR 2019.

[c] Liutkus et al., Sliced-Wasserstein Flows: Nonparametric Generative Modeling via Optimal Transport and Diffusions, ICML 2019.

The motivation of using nonlinear transformation sounds not convincing. It is indeed known that traditional SW approximation generally requires a large amount of linear transformations. The bottleneck has been overcome by some existing works (Chen et al., 2020b; Kolouri et al., 2019; Nguyen et al., 2020; Deshpande et al., 2019, 2018). Why can the suggested nonlinear transformation avoid suffering from this issue? It is also not clear why to choose Eq.(6) for the nonlinear transformation. Are there any more excellent properties of the suggested nonlinear transformation compared to the existing methods? It is also necessary to make more discussions on the application of other nonlinear transformations.

The motivation of using the additional prior-encoder is not clear to me? The introduction states that it learns an unconstrained prior distribution that matches any data manifold topology. Unfortunately, I cannot find any clear explanation about this in the proposed method part.

The evaluation is highly insufficient. The paper merely compares  the proposed method with SWAE which was published in 2018. More recent methods like [Deshpande et al. 2019, Wu et al. 2019, Liutkus et al. 2019] should be compared for a more complete study. Moreover, new generative modeling methods should be evaluated quantitatively using popular metrics like inception score, FID etc. Unfortunately, this paper does study this at all. In addition, the visual results of the proposed method seems not comparable with the state of the art on the CelebA dataset.

---

> ### Author Response · Authors · 2020-11-18
> **Expanded discussion, included informative results on manifold learning, presented timing tests and additional comparisons**
>
> We thank the referee for pointing out some important works that were
> missed. We have corrected this and provided a discussion on these
> works. Upon the referees suggestion we have made several changes to our manuscript that improve our results, present additional comparisons with literature, and better motivate our goals. We think this has significantly improved our manuscript, and hope you will take the take the time to evaluate this version.
>
> The notable changes are: (a) better communication that the goal is improved latent representation and encoding the data manifold (and not generation as was probably miscommunicated), (b) including comparions of outliers generated from EPSWAE and baseline SWAE where EPSWAE far outperforms SWAE demonstrating learning in a larger region of latent space, (c) comparison of linear vs geodesic interpolation on CelebA, which shows more realistic intermediate images through geodesic interpolation (d) inclusion of quantitaive FID scores and computational times (including comparison of computational times with other nonlinearities in literature that show that our nonlinearity is simple and cheap). (e) rewording to emphasize our motivation for chosing the specific nonlinearity - it was brought to our attention that the nonlinearity is closely related to the generalized sliced wasserstein distance in Kolouri et al 2019 (we clarify this, cite a bunch of related literature, present a comparison between several equivalent nonlinearities, and discuss the motivation for choosing the specific form of nonlinearity relating to boundary considerations).
> Several smaller changes, and additional figures have been included in the Appendix.
>
> It is worth highlighting a point that may have been miscommunicated:
> The goal of the paper is in generating a richer prior, and consequently a better latent representation that encodes geometric and topological properties of the data manifold. It is important to note that one can have good generation, despite learning a poor latent representation (and vice-versa), so generation results and scores like FID may not be the best way to evaluate the latent structure. Plotting the exact form of the prior (Fig.2) and interpolation which navigates the entire latent space (fig. 5) are reasonable measures of whether the structure of latent space is learning an embedding.
> We have reworded the text to draw motivation away from generation and toward exploring learning latent embeddings.
>
>
> We provide a point by point response below (with R denoting the referees comments and A denoting our response):
>
> R: the motivation of using nonlinear transformation sounds not convincing…
>
> A:  We have included a detailed discussion on the motivation behind our choice of nonlinearity, and comparisons with others in literature in the intro, background, section 3.1 and appendix A. The gist is as follows:
> In fact, it has been brought to our attention that the sine-based non-linearity is almost an example of a generalised Radon transform discussed in Kolouri et. al. 2019 which discusses polynomial non-linearities. The two concepts are not quite the same (for example, our nonlinear transformations do not satisfy their condition H2). We chose our nonlinearity with the following motivations. A bounded non-linearity would be beneficial as unbounded non-linearity (such as cubic polynomials) have a pronounced deformation on the tails of a measure and may excessively weight outliers, which may be undesirable.
>
> The cubic (or higher order) nonlinearities also require O(L^3) flops where L is the number of latent dimensions, which is much more expensive, and not scalable to higher dimensional datasets. A sigmoid is another potential candidate, but it saturates at high values and we want the non-linearity to have an effect everywhere. We also wanted a way to easily adapt the nonlinearity to the distribution (done by choosing the frequency that are most likely to produce deformations which highlight differences in the measures). A sine was the natural choice from a mathematical view-point. However, in practice, we do not expect a significant difference between reasonable, sufficiently rich choices of nonlinearity. Kolouri et. al. 2019 offers a rich discussion on the closely related generalised SWD are useful and compares several possibilities and against max-SWD approaches. We ran comparisons on our spiral data set and found that the cubic and quintic GSWD (Kolouri et al 2019) have essentially the same effect as the sine (although do not generalise to higher dimensional data sets).
>
> We have restructured the our paper to do the following: (1) provide the above theoretical motivation for the choice of nonlinearity; (2) provide a more clear discussion about the existing work regarding improvements to SWD, such as the GSWD; (3) clearly state that using a nonlinear transformation is not itself a novel contribution; (4) provide comparisons with the cubic and quintic GSWD discussed in Kolouri et al. 2019

---

> > ### Author Response · Authors · 2020-11-18
> > **Point by point response**
> >
> > R: The motivation of using the additional prior-encoder is not clear..
> > A: We apologize for the lack of clarity. In fact, the prior-encoder is the most important feature of the model itself; the nonlinear SWD and FSC term in the loss are significantly less important. The other main contribution is the network-geodesic algorithm.
> >
> > The purpose of the method was to learn a natural latent embedding of the data which retained geometrical and topological information about the data manifold. Manifold embedding and learning representations of high dimensional data is an important problem. This cannot be done effectively with a Gaussian-based prior (which is constrained to Gaussian form). In contrast, the prior encoder allows us to learn any shape of the data manifold (for e.g. spiral in Fig. 2), thus the latent space encodes the geometry and structural properties of the data. KL divergences are intractable for arbitrary distributions, and hence one must revert to measures such as SW distance.
> >
> > The latent structure now has a manifold embedding of the data. Our network-geodesic algorithm is designed to take advantage of this by tending to interpolate through geodesics on this manifold (through regions of higher probability density). This helps ensure that intermediate faces are all realistic.
> >
> > R: The evaluation is highly insufficient.
> > A: We want to emphasise that the primary goal of the paper is in learning a prior distribution that encodes geometry and topology of the data manifold and consequently a better latent embedding. This can be naturally tested through interpolation, which navigates and samples from the latent space, and hence is a reasonable measure of whether the structure of latent space is realistic.
> > Necessarily, a structured latent space improves Interpolative power, but not necessarily generative power. Our goal is not improved generation and so we haven’t shown comparisons with the latest state-of-the-art generative papers.
> >
> > Ideas such as ‘interpolation quality’ and ‘prior quality’ are difficult to measure quantitatively. Only a few papers (with the benchmark being SWAE) are able to have non-standard priors (non-Gaussian-based) to our knowledge (and we have mentioned these). Nevertheless, we have included qualitative comparisons with several papers (on manifold learning etc., including the ones you mention, which are great at generation and do not study manifold embedding) in the intro and background.
> > Based on your feedback, we have also restructured the text to more clearly convey our goals, and included results that demonstrate the success of the prior encoder in encoding latent space.
> >
> > It is possible to have good generation despite learning a poor latent space, so scores like FID may not be the best way to evaluate the advantages of the method. However, based on your feedback we have included FID scores (we should emphasize again that our models are small and unsophisticated compared with state of the art generative models.) in Table 2 which suggests that the prior-encoder alone (EPSWAE-bl) improves generation in SWAE, with all other features of the method being equal. We also provide computational cost comparisons with Kolouri et. al. 2019

---

### Official Review · AnonReviewer3 · 2020-10-28
**Richer Priors and Geodesic Interpolation Might Improve Image Generation in VAEs**

**Rating:** 7
**Confidence:** 4

**Review:**

# Paper Summary

The paper extends the variational autoencoder framework with a richer prior distribution to model more complex correlations in the latent variable distribution. They start with a Gaussian mixture distribution as the prior for the latent variables, and add an encoder network to allow richer correlation structure in the latent variables.  Training the prior distribution requires an optimization between the prior distribution and the latent encoded distribution of the training data set. The paper starts with an existing method of optimizing the prior by computing an approximation of the Wasserstein distance between prior and encoded training distribution that uses an average over slices through the prior and encoded training distribution. The paper replaces linear projections used in prior work with a non-linear projection. The paper also employs a structural consistency term which has been used in prior work, however, the paper employs this term differently than prior work by applying it between encoder features and latent variables rather than inputs and latent variables. Since the latent variable space is now a complex and possibly a nonconvex submanifold, points in the latent space R^D  lying between points corresponding to training data points may not actually fall in the training distribution. The paper therefore proposes a method of interpolating between points in the manifold by constructing a graph between points sampled from the manifold and then choosing points lying along lines in the graph. The paper tests the method on three datasets, a synthetic 40 dimensional spiral dataset, the venerable MNIST dataset and a scaled down CELEB A dataset. Plots of the latent space trained on the spiral dataset shows that the latent space can in fact have complex internal structure.

# Pros and Cons

Improving the ability of generative models to capture high-dimensional empirical distributions accurately is a key problem in machine learning and central to the representation learning theme of the ICLR community.

The paper clearly states contributions up front, namely: using an encoder to generate richer priors for the latent variable distribution, non-linear projections for sliced Wasserstein approximation, and a graph based interpolation method.  They also alter the structural consistency term so that it applies to more abstract features instead of inputs.

The paper does a thorough and clear job of covering prior work and technical background such as the Wasserstein metric, perhaps even excessively so.

The particular choice of a sinusoidal non-linear projection, a key contribution according to the paper, is not motivated in the text. On first glance, the sinusoidal term seems like an odd choice for a non-linearity. After looking at the results which include a test on spirals, it is clearer why this might have be chosen, but is a sinusoidal term likely to be helpful for non periodic data? It might be possible to shed some light on this by investigating whether the coefficients in the non-linear term, zeta and gamma, are significantly different from zero after training on MNIST or the CELEB A data set used in the paper.

Figure 3 comparing EPSWAE and SWAE doesn’t clearly illustrate the benefits of the EP component. In the print version, both grids of images seem blurry and prone to oddities such as overly large and dark eyes. Even when one blows up the image to 3X size using the digital version the advantage does not jump out. While I recognize that evaluating generative models is hard, the observations do not clearly support the author's hypothesis that the EP component provides an advantage. Maybe they could evaluate Freschet Inception Distance over the whole data set? Use blind human reviewers to choose between EPSWAE and SWAE based on realism and report preference scores? Interestingly, there is one duplicate image in the EPSWAE grid: row 1, col 5 and row 5, col 8 look identical. Seems odd to get identical images: is this because of sampling from a discrete graph structure? The highly similar but not identical images row 1, col 1 and row 1, col 8 are more what I would expect. Maybe the advantage could be made clearer by helping us focus on relevant features. For instance, it might be that EPSWAE is a little bit less likely to generate large black eyes? I can't tell from this small sample, but a grid that focuses on this might make the point. Hair versus background also seems to be a challenge. SWAE image row 3, col 4 seems to have two hair regions for the same face, but, EPSWAE images such as row 4, col 7 also look like they have two distinct hair regions. There are a couple of SWAE images that seem particularly ill formed: row 5, col 5  and maybe row 5, col 8, and row 7, col 4. It might be worth focusing on a few of the worst examples from both EPSWAE and SWAE to show differences in tails rather than the mean? I wonder if you could do a leave-one-out kind of analysis where you check the probability of held out training data points under the prior for EPSWAE vs. SWAE to assess if the prior is capturing the empirical distribution better? You would have to invert the prior network with gradient descent to do this ... but it might work.

It might make sense to compress some of the tutorial material up front to make room for more results demonstrating the efficacy of the model. The MNIST results, fig 7 in appendix D, shows some advantage for EPSWAE: For instance, figure 7b and 7c both have more bloated numbers … especially 8, 3 and 0.  Also figure 7b has one degenerate number in position 1,2 … possibly a 2?

The paper argues that the structured latent space improves generative power. Is there any evidence of structure in the latent space after training on CELEB A? If we plot 2 or 3D projections, or plot projections of structure preservering factorizations such as PCA, do we see structure in the encoding training data points or are they distributed independently? One might also try independence tests between variables in the encoded latent space to see if pairs of variables are being encoded with correlations. This could be compared easily between EPSWAE and SWAE.

Figure 4 in section 5.4 on interpolation shows smooth interpolation between two points A and B in latent space presumably drawn from training data. The interpolations are pretty smooth. Nice! However, the authors claim is that the graph embedding gives better interpolations than linear interpolation between points in the latent space. To make this point, we would also need to see interpolations between points using linear interpolations. The plots on spiral might make this point, but it is not clear.

The ability of VAE’s to disentangle the dimensions of an empirical distribution into indpendent latent variables is sometimes seen as a feature, not a bug. For instance, if the training data truly lie along a spiral, isn’t this really a 1D latent space and not a 3D space? While I can see the appeal of improving generated distribution realism, some discussion by the paper on the merits of improving encoder and decoder vs. complexifying the latent space would help to motivate this approach.

It isn't clear to me that the structural consistency term is a good idea in general. Ideally we want the latent space to capture something fundamental about the underlying structure of the data and not features of the input. Moving the structural consistency from input to more abstract features addresses this concern somewhat, but aren't the latent values themselves the ultimate goal?

# Recommendation

 I recommend a rejection of the paper. The hypotheses (that richer priors and geodesic interpolation generate better images on realistic images) are not clearly supported by the experimental results provided.

# Questions

 For the spiral training, what was the dimensionality of the latent space? Was it 3D?

Figure 4 caption contains statement "through an intermediate sample corresponding to the midpoint in latent space". Are you actually literally using the midpoint? I thought graph embedding was being used to avoid using midpoints? Maybe the sentence is just ambiguous?

How many samples are used in the Wasserstein approximation? How were the coefficients in the multi-term loss function defined (alpha, beta and kappa)? Oh - I see these are in the appendix... Seems like appendices are becoming pretty integral to papers these days ... Probably worth including these for the main results in section 5 for the two results presented.

Step 3 in the graph embedding didn't make sense to me after reading it a couple of times. It wasn't quite clear how this sample specific weighting works. It would probably be worth expanding this a bit at the expense of background material, as it is one of the contributions of the paper.

# Other Feedback

Page 2 "Adversarial methods are harder to train" ... also adversarial methods are implicit distributions -- you can sample from them but you cannot easily calculate the likelihood of an image under the adversarial model (although you can use gradient descent to try to find the latent parameters). This makes things like outlier detection difficult.

Page 7, Section 5.2, last sentence refers to Fig 6, but I think this should be Appendix D, Fig 7?

If you flipped the name around from EPSWAE to SWEP-AE, you would have a much more memorable acronym for people to take away from your paper/talk/poster although I recognize this doesn’t have the same “build” on previous work dynamic.

---

> ### Author Response · Authors · 2020-11-18
> **Elucidate goals more clearly, updated discussion on NSW distance, added informative results on manifold learning and quantitative comparisons + timing tests. Thank you!**
>
> Thanks for your very insightful feedback and positive evaluation of the importance of this work and the general methods in this paper. We have gone over each of your comments carefully (some of the suggestions were quite creative), and made several changes to our manuscript based on your advice, which significantly improved some of our results. We think this has significantly improved our manuscript, and hope you will take the take the time to evaluate this version.
>
> The notable changes are: (a) better communication that the goal is improved latent representation and encoding the data manifold (and not generation as was probably miscommunicated), (b) including comparions of outliers generated from EPSWAE and baseline SWAE where EPSWAE far outperforms SWAE demonstrating learning in a larger region of latent space, (c) comparison of linear vs geodesic interpolation on CelebA, which shows more realistic intermediate images through geodesic interpolation (d) inclusion of quantitaive FID scores and computational times (including comparison of computational times with other nonlinearities in literature that show that our nonlinearity is simple and cheap). (e) rewording to emphasize our motivation for chosing the specific nonlinearity - it was brought to our attention that the nonlinearity is closely related to the generalized sliced wasserstein distance in Kolouri et al 2019 (we clarify this, cite a bunch of related literature, present a comparison between several equivalent nonlinearities, and discuss the motivation for choosing the specific form of nonlinearity relating to boundary considerations).
>
> Several smaller changes, and additional figures have been included in the Appendix.
> It is worth highlighting a point that may have been miscommunicated:
> The referee is correct in identifying that the goal of the paper is in generating a richer prior, and consequently a better latent generation. Plotting the exact form of the prior (Fig.2) and interpolation which navigates the entire latent space (fig. 5) are reasonable measures of whether the structure of latent space is realistic. Our goals are not to show particularly amazing generation (our models are much too small for that anyway) - and while improved latent space representations may improve generation, they are neither necessary nor sufficient to do so. We have reworded the text to draw motivation away from generation and to clarify this point; there exists many wonderful papers that do this with very high quality. Upon the referees suggestion we have also changed to Fig 3 - opting instead for a plot of the tails - where differences in latent structure are much more easily discernible.
>
> We offer a point-by-point response below (where R denotes the referees comment and A denotes our response):

---

> > ### Author Response · Authors · 2020-11-18
> > **Point by point response**
> >
> > R: The particular choice of a sinusoidal non-linear projection…
> > A:  We have included a detailed discussion on the motivation behind our choice of nonlinearity, and comparisons with others in literature in the intro, background, section 3.1 and appendix A. The gist is as follows:
> > In fact, it has been brought to our attention that the sine-basednon-linearity is almost an example of a generalised Radon transform discussed in Kolouri et. al. 2019 which discusses polynomial non-linearities. The two concepts are not quite the same (for example,
> > our nonlinear transformations do not satisfy their condition H2). We chose our nonlinearity with the following motivations. A bounded non-linearity would be beneficial as unbounded non-linearity (such as cubic polynomials) have a pronounced deformation on the tails of a measure and may excessively weight outliers, which may be undesirable.
> > The cubic (or higher order) nonlinearities also require O(L^3) flops where L is the
> > number of latent dimensions, which is much more expensive, and not scalable to higher dimensional datasets. A sigmoid is another potential candidate, but it saturates at high values and we want the non-linearity to have an effect everywhere. We also wanted a
> > way to easily adapt the nonlinearity to the distribution in a straightforward and efficient way (done by choosing the frequency that are most likely to produce deformations which highlight differences in the measures). We also wanted a choice that would adapt easily to high dimensional data sets. A sine was the natural choice from a mathematical view-point. However, in practice, we do not expect a significant difference between reasonable, sufficiently rich choices of nonlinearity. Kolouri et. al. 2019 offers a rich discussion on the closely related generalised SWD are useful and compares several possibilities and against max-SWD approaches. We have done some studies on our spiral data set and found
> > that the cubic and quintic GSWD discussed in Kolouri et al have essentially the same effect as the sine (although do not generalise as easily to higher dimensional data sets).
> >
> > We have restructured the intro, background, and sections 3.1 and appendix A (both relating to nonlinear SW distances) in our paper to do the following: (1) provide the above theoretical motivation for the choice of nonlinearity; (2) provide a more clear discussion about the existing work regarding improvements to SWD, such as the GSWD; (3) clearly state that using a nonlinear transformation is not itself a novel contribution; (4) provide comparisons with the cubic and quintic GSWD discussed in Kolouri et al. 2019
> >
> > R: Figure 3 comparing EPSWAE and SWAE…
> > A: The ‘goodness’ of generative models is difficult, as the referee correctly pointed out. However, the goal for us, was really to learn a better prior (which EP is successful can do in a way that isn’t possible with conventional AE priors). The easy-to-visualise spiral dataset really drives this message home.
> >
> > It is possible to have good generation, despite learning a poor latent space, so scores like FID may not be the best way to evaluate the advantages of the method. Nevertheless, we have included FID scores in Table 2, including not only SWAE vs EPSWAE but also the FID scores involving the same model and hyperparameters trained (A) with standard SWD and no FSC term; (B) with nonlinear SWD and no FSC term; (C) with standard SWD and FSC term; and (D) both nonlinear SWD and FSC term. The latter has the lowest FID score, whereas the (A-C) are all close. There is a big jump between any of the EPSWAE variants and SWAE. These results suggest that our improved latent representation is leading to improved generation according to the FID score. We should emphasize again that our models are small and unsophisticated compared with state of the art generative models.
> >
> > The points about showing EPSWAE vs SWAE comparisons that focus on a
> > specific feature or the tails are well taken and we have included Figure 3 that studies the tails. This may be a better metric (than simple generation) for understanding how well the latent space encodes information. Naturally both methods generate poorer images when studying extreme outliers, but it is interesting to see that the break down is quite different (and we believe EPSWAE fails more gracefully, suffering from some mode collapse, rather than the more catastrophic behavior of SWAE).

---

> > > ### Author Response · Authors · 2020-11-18
> > > **Point by point response**
> > >
> > > R: It might make sense to compress some of the tutorial material…
> > > A: We’ve incorporated your suggestion (moved the sliced wasserstein proof to appendix) and highlighted MNIST in the main text as well!
> > >
> > > R: The paper argues that the structured latent space improves generative power…
> > > A: I think it’s important to highlight a point that seems to have been poorly conveyed: A structured latent space improves interpolative power, but not necessarily generative power.The paper tries to learn a prior that encodes a latent embedding that retains geometric information of the data. Based on your feedback, we have restructured the text to make this more clear, and highlighted the results that are indicative of how well the latent space is encoding information.
> > > Now, do we see structure in the encoding training data? We have added several citations (Lu et al `98, Wei et al `16, Rahimi et al `05) which study image and video manifolds and support the point that manifold structure is inherently present in many datasets. The
> > > smoothness of geodesic-interpolations also suggests such structure is present also in the latent embedding.
> > >
> > > R: Figure 4 in section 5.4 on interpolation shows smooth interpolation...
> > > A: Thanks! A graph interpolation is a much more natural way of interpolating along manifolds (as you may imagine). We have included comparisons also with linear interpolation for CelebA in Fig 4 (a) ! The referee pointed out that the spiral is also the perfect dataset to demonstrate that -  we’ve highlighted this in the spiral in Fig. 2.
> > >
> > > R: The ability of VAE’s to disentangle the dimensions of an empirical distribution into indpendent latent variables…
> > > A: Absolutely, the spiral data does lie on a 1D manifold.  The input was originally a 3D spiral mapped forward under a random linear transformation into a 40D space (with additional noise added at the end). We use a 3D latent space. We see that EPSWAE learns the something approximating the original 3D geometry. It may be helpful to think of this in terms of geometric measure theory - we want to learn a prior distribution  that maintains the original geometry, for example, a distribution that is likely to be approximately
> > > concentrated along a sub-manifold.  All this can be solved ad hoc if you know the exact dimensionality of the manifold. However, if you don’t know the inherent dimensionality, as is true in a lot of real data, in this method, the geometry is naturally encoded as long as the
> > > latent dimensionality is large enough (>=3 in the spiral case).
> > >
> > > R. It isn't clear to me that the structural consistency term …
> > > A: This is a valid concern, and one we had initially as well. Sainburg et al `18 presents a good discussion on why any type of structural consistency may be useful for interpolation, and they found minor improvements upon inclusion. As you correctly pointed out, we want to extract underlying geometric and topological properties of the data. The abstract features, one would hope may already be capturing some aspects of the underlying data structure, so while this term may not help generation, feature-isometry in latent space should help preserve geometry.
> > >
> > >
> > > Now, onto the questions.
> > >
> > > 1) Spiral training latent space dimensionality was 3D.
> > >
> > > 2) This is a purely implementational detail. With enough prior samples, the network-geodesic could generate a very smooth path. However, one requires high samples to fill in a high dimensional latent space, hence we found it sometimes (in Appendix H) necessary to ‘fill in’ with midpoints. We've included a discussion on this in Appendix H.  It's worth noting that even just one or two points selected by the network-geodesic can result in completely different (from linear) interpolation. (e.g. Fig. 1 interpolation A-C-B vs (linear)A-B).
> > >
> > > 3) Yes, these details are in the Appendix. Real estate in the main text is expensive these days. :) Your suggestion makes sense, we have moved some of this to the main text.
> > >
> > > 4) We’ve tried to reword to make this bullet point to make it clear. It’s easier to consider  h=1. Basically, nearby (closer than some threshold) samples are connected forming traversable paths. This encourages shorter hops along the paths. The threshold is proportional to the degree of the sample (a measure of how central it is). Thus, high degree/more central samples, have high thresholds (and so, on average, more paths go through them). In the context of a manifold, this corresponds to a geodesic looking for the shortest path through the best approximation of the manifold. Though it is worth mentioning that data isn't just a manifold, it is a full probability distribution, which is significantly more general. The network is designed so that network-geodesics tend to go through points where the probability density is relatively high. We've reworded this bullet point for clarity.
> > >
> > > 5) Other feedback: good insight. Also, nice suggestion for the name (but doesn’t quite roll of the tongue like EPSWAE, pronounced epsway)

---

> > ### Comment · AnonReviewer3 · 2020-11-20
> > **Significantly stronger paper demonstrating better interpolation with proposed approach.**
> >
> > The paper is significantly stronger after this revision. Figure 2 makes the linear vs. geodesic interpolation clear. Figure 3 shows a clear difference between EPSWAE and SWAE. EPSWAE produces better images at higher absolute levels of noise ... I don't know if this suggests better generalization or just rescaling of noise...  However, unlike the previous version of this figure, it does show distinct behavior between the approaches and raises some interesting questions about what is going on. Figure 4 is also much stronger. The geodesic interpolation clearly stays more within a realistic space than the linear interpolation. The notion of generating "a better representation" is good, but what does better mean? The idea of focusing on interpolation is interesting, but Figure 3 doesn't speak to this line of argument.  The difference between linear and geodesic interpolation in Figure 4 does suggest that some non-trivial nonconvex substructure has been captured, otherwise linear interpolation would work just as well as geodesic.  The FID scores also underscore that the representations generated by the proposed approach are different and that all of the components in the ablation study are contributing something. I think there is now much stronger evidence now that you have found something interesting.

---

> > > ### Author Response · Authors · 2020-11-23
> > > **Thanks for your positive review and clarification of 'better representation'**
> > >
> > > We thank you for acknowledging the improvements to our paper, and your positive review of our results. We appreciate your comments that have helped significantly improve the manuscript. We are glad that you appreciate our improved discussions, our new FID scores , ablation study, and agree that the new Fig.3 and Fig. 4 shows significant improvements of EPSWAE and geodesic interpolation.
> > >
> > > A minor comment in response to your question: the idea of better latent representation is indeed vague by definition, and are along the lines of what you state in your comment. There are several works (cited in the paper) that speak to the manifold hypothesis, i.e., real world data can be thought of as lying on a (approximate) manifold. We primarily think of a 'better representation' as a better way to capture the nontrivial data manifold, i.e., encoding both the geometry and topology of the data, allowing realistic interpolation to occur ON the data manifold (as seen in Fig. 4). In our opinion, this is a very important question in learning representations. With our method, typically larger regions in latent space (wrt prior measure rather than Lebesgue measure) will see training (Figure 3 speaks to this aspect), and interpolations will go through fewer untrained regions/holes (Figure 4 speaks to this aspect) in latent space since natural manifolds are locally smooth without many singularities/holes. The intro and relevant results sections outline the above in some detail.
> > >
> > > Thanks a lot for all your feedback, which have been very useful. We hope you will agree that the strength of the underlying theoretical ideas and our much improved validation make this paper a good fit and now acceptable for publication. We'd appreciate it if you could update your score to reflect your new assessment.

---

### Comment · Area_Chair1 · 2020-11-22
**Time for discussion**

Dear Reviewers,

The authors have provided a detailed response and uploaded their revised manuscript. Would you please take a careful look at their response and revision? Please respond to the authors and update your review accordingly.

Thanks,
AC

---

### Decision · Program_Chairs · 2021-01-07
**Final Decision**

**Decision:**

Reject

**Comment:**

All three referees have provided detailed comments, both before and after the author response period. While the authors have carefully revised the paper and provided detailed responses, leading to clearly improved clarity and quality, there remain clear concerns on novelty (at least not sufficiently supported with ablation study) and experiments (neither strong enough nor sufficient to support the main hypotheses). The authors are encouraged to further improve their paper for a future submission.